# Safe Decision Transformer with Learning-based Constraints

**Ruhan Wang**
Indiana University
ruhwang@iu.edu

**Dongruo Zhou**
Indiana University
dz13@iu.edu

## Abstract

In the field of safe offline reinforcement learning (RL), the objective is to utilize offline data to train a policy that maximizes long-term rewards while adhering to safety constraints. Recent work, such as the Constrained Decision Transformer (CDT) [30], has utilized the Transformer [38] architecture to build a safe RL agent that is capable of dynamically adjusting the balance between safety and task rewards. However, it often lacks the stitching ability to output policies that are better than those existing in the offline dataset, similar to other Transformer-based RL agents like the Decision Transformer (DT) [7]. We introduce the Constrained Q-learning Decision Transformer (CQDT) to address this issue. At the core of our approach is a novel trajectory relabeling scheme that utilizes learned value functions, with careful consideration of the trade-off between safety and cumulative rewards. Experimental results show that our proposed algorithm outperforms several baselines across a variety of safe offline RL benchmarks.

## 1  Introduction

Recent studies have demonstrated the Transformer's [38] state-of-the-art performance across a range of applications, including natural language processing [40, 4] and computer vision [27, 3]. When applied to the domain of Reinforcement Learning (RL), a recent trend is to treat the decision-making problem as a sequence modeling problem, using auto regressive models such as Transformer which maps the history information directly to the action [7] or the next state [18]. Notably, the Decision Transformer (DT) [7] effectively extends the Transformer architecture to offline RL tasks, showcasing strong performance, particularly in sequential modeling. However, it is worth noting that while DT excels in maximizing long-term rewards, it may not always align with the complexities of real-world tasks. In practice, many tasks cannot be simplified solely into optimizing a single scalar reward function, and the presence of various constraints significantly narrows the spectrum of feasible solutions [12]. Such a setting is called safe RL, which has been studied in lots of safety-related decision-making problems. For instance, it is crucial that robots interacting with humans in human-machine environments prioritize human safety and avoid causing harm. In the realm of recommender systems, it is imperative to avoid recommending false or racially discriminatory information to users. Similarly, when self-driving cars operate in real-world environments, ensuring safety is paramount [36, 14, 31].

Constrained Decision Transformer (CDT) [30] serves as a pioneering work which extends the Transformer-based RL to the safe RL regime, which builds upon the foundation of the DT while incorporating essential safety constraints. CDT's core objective is to acquire a safe policy from an offline dataset. Its distinguishing feature is the ability to maintain zero-shot adaptability across a range of constraint thresholds, rendering it highly suitable for real-world reinforcement learning applications burdened by constraints. While CDT demonstrates exceptional competitiveness in safe offline reinforcement learning tasks, it shares a common limitation with DT—an absence of the 'stitching' capability. This crucial ability involves integrating sub-optimal trajectory segments to form

Submitted to 38th Conference on Neural Information Processing Systems (NeurIPS 2024). Do not distribute.

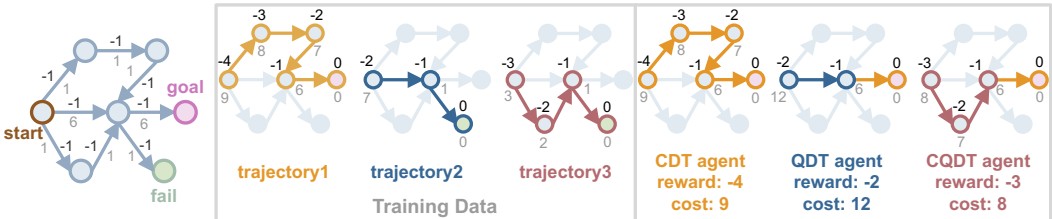

**Figure 1:** The toy example is presented where arrows denote the node connections. Here we use black number to denote rewards and gray nodes to denote costs. The demonstration example shows that cost return-based method (CDT) fails to find the shortest path to the goal since it lacks the stitching ability. In contrast, Q-learning-based DT (QDT) finds the shortest path, while it violates the safety constraints. Our proposed CQDT enjoys the superiority of both methods.

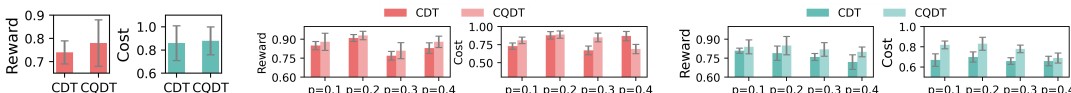

**Figure 2:** Performance comparison between CDT and CQDT, with results averaged over three random seeds.

**Figure 3:** Stitching ability comparison between CDT and CQDT on reward-suboptimal datasets with results averaged over three random seeds.

**Figure 4:** Stitching ability comparison between CDT and CQDT on cost-suboptimal datasets with results averaged over three random seeds.

a near-optimal trajectory, a pivotal characteristic highly desired in offline reinforcement learning agents. What is more challenging is that for RL with safety constraints, the agent not only needs to stitch trajectories to achieve better reward feedback but also needs to guarantee that the obtained policy is still *safe* after stitching, not being affected by unsafe trajectories in the offline dataset. Thus, we raise the following question:

**Can we design DT-based algorithms that output safe policies with the stitching ability?**

We answer this question affirmatively. To better demonstrate our algorithm design, we propose a toy example involving finding the path that maximizes reward under cost constraints, as illustrated in Figure 1. The task's objective is to identify a path with the highest reward while adhering to a cost constraint (cost limitation = 10). The training data covers segments of the optimal path, but none of the training data trajectories encompass the entire optimal path. The agent must synthesize these fragmented trajectories to determine the optimal path within the cost constraint. For the existing DT-based RL algorithms such as Q-learning Decision Transformer (QDT) [42], they are able to stitch suboptimal trajectories into the optimal ones, but they are unable to maintain safety during the stitching phase. For the existing DT-based safe RL methods such as CDT, they can only obtain suboptimal policies while satisfying the safety guarantee, due to the lack of stitching ability. Thus, we propose the **Constrained Q-learning Decision Transformer (CQDT)** method to address the issues in existing methods, as shown in Figure 1. The main contributions are listed as follows.

- CQDT seeks to improve the quality of the training dataset by utilizing cost and reward critic functions from the Constraints Penalized Q-learning framework [41] to relabel the return-to-go (RTG) and cost-to-go (CTG) values in the training trajectories. This relabeled dataset is subsequently used to train a Decision Transformer-based safe RL method, such as CDT.

- We provide a comparative analysis of our CQDT against various safe RL benchmarks across different RL task settings. The results are summarized in Figure 2, demonstrate that CQDT consistently outperforms all existing benchmarks in several offline safe RL tasks.

- We also show that our proposed CQDT enjoys better stitching ability compared with CDT, which suggests that CQDT can better utilize suboptimal trajectories in the offline dataset. The results are summarized in Figure 3 and Figure 4.

## 2 Related Work

**Offline Reinforcement Learning.** Offline Reinforcement Learning refers to a data-driven approach to the classic RL problem [25]. It focuses on deriving policies from pre-existing data without

requiring further interaction with the environment. The practical applications of offline RL are extensive, spanning domains such as healthcare [13] and the acquisition of robotic manipulation skills [19]. However, its inherent challenge lies in the potential disparity between the learned policy and the behavior that generated the initial offline data [21]. Addressing this challenge has spurred the development of various methodologies, including constraining the learned policy close from the behavior policy [11, 21, 22, 44, 17, 20, 39, 32, 18]. There is a recent line of work aiming at providing a Transformer-based policy without explicitly constraining the distribution shift issue [7]. Our work falls into that regime, which uses a Transformer as the policy model focusing on the safe RL setting.

**Offline Safe Reinforcement Learning.** Offline Safe Reinforcement Learning aims to acquire constrained policies from pre-collected datasets, ensuring adherence to safety requirements throughout the learning process. This approach amalgamates techniques from both offline RL and safe RL, leveraging methodologies from both domains [23]. Certain methods tackle the constrained optimization problem through stationary distribution correction, employing Lagrangian constraints to ensure safe learning [41, 33]. Our work does not take the Lagrangian approach to learn a safe policy. Instead, our method explicitly treats the safe policy learning as a sequence modeling problem, similar to the previous CDT approach [30]. Such an approach enjoys the simplicity regarding the algorithm design, as well as the robustness to the algorithm performance.

## 3  Preliminaries

**Safe Offline RL.** We formulate the environment as a Constrained Markov Decision Process (CMDP), a mathematical framework for addressing the safe RL problem [2]. A CMDP comprises a tuple $(\mathcal{S}, \mathcal{A}, \mathcal{P}, r, c, \mu_0)$, where $\mathcal{S}$ denotes the state space, $\mathcal{A}$ signifies the action space, $\mathcal{P} : \mathcal{S} \times \mathcal{A} \times \mathcal{S} \to [0, 1]$ represents the transition function, $r : \mathcal{S} \times \mathcal{A} \times \mathcal{S} \to \mathbb{R}$ stands for the reward function, and $\mu_0 : \mathcal{S} \to [0, 1]$ indicates the initial state distribution. In CMDP, $c : \mathcal{S} \times \mathcal{A} \times \mathcal{S} \to [0, C_{\max}]$ quantifies the cost incurred for violating constraints, where $C_{\max}$ denotes the maximum cost allowable.

We denote the policy by $\pi : \mathcal{S} \times \mathcal{A} \to [0, 1]$, while $\tau = \{s_1, a_1, r_1, c_1, \ldots, s_T, a_T, r_T, c_T\}$ delineates the trajectory containing state, action, reward, and cost information throughout the maximum episode length $T$. We use $\tau.s_t, \tau.a_t, \tau.r_t, \tau.c_t$ to denote the $t$-th state, action, reward and cost in trajectory $\tau$. For each time step $t$, the action $a_t$ is drawn following the distribution $\pi(s_t, \cdot)$, and the next state $s_{t+1}$ is drawn following the transition function $\mathcal{P}(s_t, a_t, \cdot)$. The cumulative reward and cost for a trajectory $\tau$ are represented as $R(\tau) = \sum_{t=1}^{T} r_t$ and $C(\tau) = \sum_{t=1}^{T} c_t$. We also denote $R_t = \sum_{i=t}^{T} r_i$ by the return-to-go (RTG) at $t$-th step, $C_t = \sum_{i=t}^{T} c_i$ by the cost-to-go (CTG) at $t$-th step as well. For simplicity, we define $Q_r^\pi(s, a) = \mathbb{E}_{\tau \sim \pi}[R(\tau) | \tau.s_1 = s, \tau.a_1 = a]$ as the expected return of the policy starting from initial state $s$ and action $a$. Similarly, we denote $Q_c^\pi(s, a) = \mathbb{E}_{\tau \sim \pi}[C(\tau) | \tau.s_1 = s, \tau.a_1 = a]$ as the expected cost. The agent's objective is to determine a policy $\pi^\kappa$ that maximizes the reward while ensuring that the cumulative cost for constraint violation remains below the threshold $\kappa$:

$$\pi^\kappa = \arg\max_\pi \mathbb{E}_{\tau \sim \pi, \tau.s_1 \sim \mu_0}[R(\tau)],$$
$$\text{s.t. } \mathbb{E}_{\tau \sim \pi, \tau.s_1 \sim \mu_0}[C(\tau)] \leq \kappa. \tag{1}$$

For safe offline RL, the agent learns the optimal safe policy purely from a static dataset $\mathcal{T}$ that is previously collected with an unknown behavior policy (or policies). Specifically, $\mathcal{T}$ consists of $m$ episodes $\tau_i$ with the maximum length $T$, which is $\mathcal{T} := \{\tau_1, \ldots, \tau_m\}$.

**Constrained Decision Transformer.** Our algorithm builds on the Constrained Decision Transformer (CDT) [30]. We briefly introduce the details of CDT here, and we leave more details in the appendix. CDT utilizes the return-conditioned sequential modeling framework to accommodate varying constraint thresholds during deployment, ensuring both safety and high return. CDT employs a stochastic Gaussian policy representation to generate the action at $t$-th time step, i.e., $a_t \sim \pi_\theta(\cdot|o_t) = \mathcal{N}(\mu_\theta(o_t), \Sigma_\theta(o_t))$, where $o_t = \{R_{t-K:t}, C_{t-K:t}, s_{t-K:t}, a_{t-K:t-1}\}$ represents the truncated history from step $t - K$ to $t$, $K \in \{1, \ldots, t - 1\}$ indicates the context length and $\theta$ denotes the CDT policy parameters. During the training phase, CDT generates the training set $\{(o_t, a_t)\}$ by splitting trajectories in $\mathcal{T}$ into shorter contexts with length $K$, then it trains the policy by minimizing the prediction loss between $a_t$ and $\pi_\theta(\cdot|o_t)$. During the inference phase, CDT selects the initial return-to-go $R_1$ as well as the cost-to-go $C_1$, then it selects the action $a_t$ based on the current

---
**Algorithm 1** Relabeling trajectory set
---
**Require:** Trajectories dataset $\mathcal{T}$, cost limitation list $\mathcal{K} = [\kappa_1, \kappa_2, ..., \kappa_m]$, reward critic functions
    $\mathbb{Q}_r = \emptyset$, cost critic functions $\mathbb{Q}_c = \emptyset$
  1: //Run CPQ under various cost limitations to derive distinct reward critic and cost critic functions.
  2: **for** $\kappa_i$ in $\mathcal{K}$ **do**
  3:     Run CPQ(Algorithm 3) with the constraint limit $\kappa_i$, obtain the reward and cost critic $Q_r, Q_c$
  4:     Set $\mathbb{Q}_r[\kappa_i] = Q_r; \mathbb{Q}_c[\kappa_i] = Q_c$
  5: **end for**
  6: //Relabel trajectories in the dataset.
  7: **for** $\tau$ in $\mathcal{T}$ **do**
  8:     Select $\kappa^\tau = \arg\min_{\kappa_i \in \mathcal{K}} |\mathbb{Q}_c[\kappa_i](\tau.s_0, \tau.a_0) - C_0|$
  9:     Set $Q_r^\tau = \mathbb{Q}_r[\kappa^\tau]; Q_c^\tau = \mathbb{Q}_c[\kappa^\tau]$
 10:     Take $\tau'$ as the output of Algorithm 2, based on $\tau$, $Q_r^\tau$ and $Q_c^\tau$, $\mathcal{T} \leftarrow \mathcal{T} \cup \{\tau'\}$
 11: **end for**
 12: Use Pareto-Frontier augmentation method in Algorithm 4 to augment $\mathcal{T}$
**Ensure:** New trajectory dataset $\mathcal{T}$
---

$R_t, C_t$. It is worth noting that CDT enables the agent to dynamically adjust its constraint threshold during deployment without using a Lagrangian-type update.

**Constraints Penalized Q-learning.** Our algorithm relies on a component that delivers precise value estimates for the state. We take a state-of-the-art method called Constraints Penalized Q-learning (CPQ) [41] as our primary approach and discuss its details as follows. CPQ adopts the actor-critic framework, which maintains four components at each iteration. They are a policy $\pi : \mathcal{S} \times \mathcal{A} \to [0, 1]$, a discriminator $\nu : \mathcal{S} \times \mathcal{A} \to \{0, 1\}$ that decides whether a given state-action pair $(s, a)$ are out-of-distribution (OOD) of the behavior policy of the offline dataset; a reward critic function $Q_r : \mathcal{S} \times \mathcal{A} \to \mathbb{R}$ and a cost critic function $Q_c : \mathcal{S} \times \mathcal{A} \to \mathbb{R}$. During the training phase, CPQ first pre-trains $\nu$ by a Conditional Variational Autoencoder (CVAE) [16, 43]. At $t$-th step, CPQ maintains its current policy $\pi$. Then it updates the cost critic first, following

$$Q_c = \arg\min_Q -\alpha \mathbb{E}_{s,a \sim \mathcal{T}}[Q(s, a)\nu(s, a)]$$
$$+ \mathbb{E}_{s,a,s',c \sim \mathcal{T}}[(Q(s, a) - c - \gamma \mathbb{E}_{a' \sim \pi(\cdot|s')}[Q(s', a')])^2],$$

where $0 < \gamma < 1, \alpha$ are tunable parameters. Next, CPQ updates the reward critic by optimizing the following cost-penalized Bellman equation, which is

$$Q_r = \arg\min_Q \mathbb{E}_{s,a,s',r \sim \mathcal{T}}[(Q(s, a) - r$$
$$- \gamma \mathbb{E}_{a' \sim \pi}[\mathbb{I}(Q_c(s', a') \le \kappa)Q(s', a')])^2], \tag{2}$$

where $\kappa$ is the cost constraint defined in Equation (1). Finally, given $Q_r$ and $Q_c$, CPQ performs any policy optimization method to obtain $\pi'$, which maximizes the following constrained optimization problem:

$$\pi' = \arg\max_\pi \mathbb{E}_{s \sim \mathcal{T}} \mathbb{E}_{a \sim \pi(\cdot|s)}[\mathbb{I}(Q_c(s, a) \le \kappa)Q_r(s, a)]. \tag{3}$$

## 4   Method

**Algorithm Overview.** We present a framework called Constrained Q-learning Decision Transformer (CQDT), which exploits the Constrained Dynamic Programming approach, specifically CPQ, to overcome the limitations of CDT. The algorithm details are summarized in Algorithm 1. Intuitively speaking, CQDT takes a trajectory dataset $\mathcal{T}$ as its input and outputs an augmented dataset $\mathcal{T}'$ based on $\mathcal{T}$. It then trains CDT over the augmented dataset $\mathcal{T}'$. Next, we describe how to augment $\mathcal{T}$ in steps.

**First Step: Training CPQ to Obtain Value Functions**. The objective of this step is to train CPQ to obtain precise estimates of action value functions across different cost limit $\kappa$ settings. We maintain a list of cost limitations, $\mathcal{K}$, which includes $m$ different cost limits, $\kappa_1, \ldots, \kappa_m$, and run CPQ $m$ times to obtain corresponding $Q_r$ and $Q_c$ functions, denoted as $\mathbb{Q}_r[\kappa_i]$ and $\mathbb{Q}_c[\kappa_i]$, respectively, as outlined

---

**Algorithm 2** Relabeling one trajectory

---

**Require:** Trajectory $\tau$, reward critic $Q_r^\tau$ and cost critic $Q_c^\tau$
1: Set $T$ as the length of $\tau$, the new trajectory $\tau' = \tau$, $\tau'.R_{T+1} = \tau'.C_{T+1} = 0$
2: **for** $t = T + 1, \ldots, 2$ **do**
3:     Set $V_r^\tau(\tau.s_t) = Q_r^\tau(\tau.s_t, \tau.a_t)$, $V_c^\tau(\tau.s_t) = Q_c^\tau(\tau.s_t, \tau.a_t)$
4:     $\tau'.R_{t-1} \leftarrow \tau.r_{t-1} + \tau'.R_t$, $\tau'.C_{t-1} \leftarrow \tau.c_{t-1} + \tau'.C_t$
5:     **if** $V_c^\tau(\tau.s_t) \leq \tau'.C_t$ and $V_r^\tau(\tau.s_t) \geq \tau'.R_t$ **then**
6:         $\tau'.R_{t-1} \leftarrow \tau.r_{t-1} + V_r^\tau(\tau.s_t)$, $\tau'.C_{t-1} \leftarrow \tau.c_{t-1} + V_c^\tau(\tau.s_t)$
7:     **end if**
8: **end for**
**Ensure:** Relabeled trajectory $\tau'$

---

in line 4 of Algorithm 1. In the subsequent steps, the $\mathbb{Q}_r$ and $\mathbb{Q}_c$ lists are utilized to relabel the RTG and CTG for each trajectory. For additional details, please refer to Appendix A.1.

**Second Step: Relabeling Trajectories.** Now we describe how to relabel a given trajectory $\tau$ using $\mathbb{Q}_r$ and $\mathbb{Q}_c$ in detail. The steps are summarized in Algorithm 2. To demonstrate that, recall that our goal is to learn $\pi^\kappa$ that maximizes the expected return under the $\kappa$ constraint. Assume that for the trajectory $\tau$, the policy $\pi$ that generates $\tau$ satisfies the constraint $\kappa$. Therefore, in order to further push the agent to learn $\pi^\kappa$ instead of only learning $\pi$, we generate a new trajectory $\tau'$ identical to $\tau$, with different $\tau'.R_i, \tau'.C_i$, to make $\tau'$ similar to a trajectory generated by $\pi^\kappa$. Our strategy is simple: we first replace the last RTG and CTG of $\tau'$ as 0. At the $t$-th step of $\tau'$, we would like to calculate $\tau'.R_{t-1}$ and $\tau'.C_{t-1}$. Then we either to use the existing RTG ($\tau'.R_t$) and CTG ($\tau'.C_t$) to update $\tau'.R_{t-1}$ and $\tau'.C_{t-1}$ (line 4 in Algorithm 2), or we use the learned reward critic and cost critic to update $\tau'.R_{t-1}$ and $\tau'.C_{t-1}$ (line 6 in Algorithm 2), if the reward critic and cost critic provide a more "aggressive" approximation, i.e., $V_c^\tau(\tau.s_t) \leq \tau'.C_t$ and $V_r^\tau(\tau.s_t) \geq \tau'.R_t$ (line 5 in Algorithm 2). Here $V_r^\tau$ and $V_c^\tau$ are learned reward and cost critics, and they are selected from $\mathbb{Q}_r$ and $\mathbb{Q}_c$ to make sure that the cost constraint estimation $Q_c^\tau$ is close to the true CTG $C_0$ (line 8 in Algorithm 2). We summarize the relabeling process in Algorithm 2.

The most notable difference between our relabeling strategy and the previous one for offline RL [42] is that we relabel RTG and CTG *jointly and simultaneously*. If we only relabel each trajectory based on their RTG and CTG separately, we might obtain unsafe trajectories, which hurts the overall performance of CQDT. Instead, our strategy ensures that, each safe trajectory will still be safe after relabeling, which is crucial for the safe RL setting. Our experimental results in the later section suggest the effectiveness of our relabeling strategy.

**Third Step: Post-Processing Steps for the Final Trajectory.** Now, we have produced an augmented trajectory dataset $\mathcal{T}$ which consists of the original trajectories $\tau$ and the new trajectories $\tau'$. Finally, we introduce some additional post-processing steps over $\mathcal{T}$ from existing works [30, 42] for the further performance improvement of CQDT.

The first post-processing technique we adopt is to resolve the potential conflict between a high RTG and a low CTG. Due to the nature of safe RL, we would like to first guarantee the safety of our learned policy. Following [30], we use a Pareto Frontier-based data augmentation technique to further generate new trajectories and add them to $\mathcal{T}$. The second post-processing technique aims to maintain the consistency of the RTG and CTG within the input sequence of CDT. Due to space limitations, we defer the detailed in Appendix A.2.

## 5    Experiment

In this section, we begin by outlining the fundamental settings of our experiment. We then show the performance of CQDT under a series of experiments, each addressed a key challenge as follows:

- How does CQDT compare with CDT and other offline safe reinforcement learning methods in terms of performance? Additionally, how does the choice of the value function affect CQDT's performance?

- Is CQDT capable of performing effective stitching?

| | CarCircle | | CarRun | | AntRun | | DroneCircle | | DroneRun | | Average | |
|---|---|---|---|---|---|---|---|---|---|---|---|---|
| | reward | cost | reward | cost | reward | cost | reward | cost | reward | cost | reward | cost |
| BC-Safe[*] | 0.500 | 0.840 | 0.940 | 0.220 | 0.650 | 1.090 | 0.560 | 0.570 | 0.280 | 0.740 | 0.586 | 0.692 |
| BCQ-Lag[*] | 0.630 | 1.890 | 0.940 | 0.150 | 0.760 | 5.110 | 0.800 | 3.070 | 0.720 | 5.540 | 0.770 | 3.152 |
| BEAR-Lag[*] | 0.740 | 2.180 | 0.680 | 7.780 | 0.150 | 0.730 | 0.780 | 3.680 | 0.420 | 2.470 | 0.554 | 3.368 |
| COptiDICE[*] | 0.490 | 3.140 | 0.870 | 0.000 | 0.610 | 0.940 | 0.260 | 1.020 | 0.670 | 4.150 | 0.580 | 1.850 |
| CDT[*] | 0.750 | 0.950 | 0.990 | 0.650 | 0.720 | 0.910 | 0.630 | 0.980 | 0.630 | 0.790 | 0.744 | 0.856 |
| CPQ[*] | 0.710 | 0.330 | 0.950 | 1.790 | 0.030 | 0.020 | -0.220 | 1.280 | 0.330 | 3.520 | 0.360 | 1.388 |
| CQDT(ours) | $0.767_{\pm 0.081}$ | $0.895_{\pm 0.103}$ | $0.994_{\pm 0.361}$ | $0.886_{\pm 0.138}$ | $0.735_{\pm 0.094}$ | $0.832_{\pm 0.205}$ | $0.753_{\pm 0.075}$ | $0.981_{\pm 0.285}$ | $0.640_{\pm 0.075}$ | $0.812_{\pm 0.046}$ | $0.778_{\pm 0.137}$ | $0.881_{\pm 0.155}$ |
| BCQL-CDT | $0.733_{\pm 0.206}$ | $0.893_{\pm 0.002}$ | $0.983_{\pm 0.186}$ | $1.758_{0.072}$ | $0.673_{\pm 0.029}$ | $0.793_{\pm 0.038}$ | $0.685_{\pm 0.275}$ | $0.747_{\pm 0.143}$ | $0.621_{\pm 0.101}$ | $0.597_{\pm 0.239}$ | $0.739_{\pm 0.159}$ | $0.958_{\pm 0.099}$ |
| BEARL-CDT | $0.735_{\pm 0.227}$ | $0.929_{\pm 0.178}$ | $0.999_{\pm 0.059}$ | $1.707_{\pm 0.134}$ | $0.665_{\pm 0.229}$ | $0.719_{\pm 0.162}$ | $0.684_{\pm 0.034}$ | $0.766_{\pm 0.122}$ | $0.597_{\pm 0.266}$ | $0.647_{\pm 0.091}$ | $0.736_{\pm 0.163}$ | $0.953_{\pm 0.137}$ |
| Ablation① | $0.785_{\pm 0.145}$ | $0.981_{\pm 0.238}$ | $0.977_{\pm 0.296}$ | $1.357_{\pm 0.082}$ | $0.560_{\pm 0.053}$ | $0.313_{\pm 0.218}$ | $0.633_{\pm 0.076}$ | $0.837_{\pm 0.224}$ | $0.636_{\pm 0.094}$ | $0.551_{\pm 0.175}$ | $0.718_{\pm 0.133}$ | $0.808_{\pm 0.187}$ |
| Ablation② | $0.767_{\pm 0.201}$ | $0.964_{\pm 0.230}$ | $0.982_{\pm 0.186}$ | $1.901_{\pm 0.110}$ | $0.702_{\pm 0.194}$ | $0.874_{\pm 0.076}$ | $0.764_{0.263}$ | $1.202_{\pm 0.130}$ | $0.621_{\pm 0.149}$ | $0.776_{\pm 0.051}$ | $0.767_{\pm 0.198}$ | $1.143_{\pm 0.119}$ |

**Table 1:** Evaluation results for normalized reward and cost are provided. **Bold**: Safe agents with a normalized cost smaller than 1. Gray: Unsafe agents. **Blue**: Safe agent achieving the highest reward. Each value is averaged over 3 distinct cost thresholds ($\kappa = 10, 20, 40$), 20 episodes and 3 seeds, following the setting in [29]. Results for baselines with $*$ are copied from [29]. BCQL-CDT: Validation of the impact of the value function on algorithm performance: Using the value functions in BCQ-Lag. BEARL-CDT: Validation of the impact of the value function on algorithm performance: Using the value functions in BEAR-Lag. Ablation①: Effect of removing constraints learning by separately considering reward and cost relabeling. Ablation②: Without PF-based augmentation.

- What is the impact of the various components of CQDT on its overall performance?

- How does CQDT perform in a sparse reward environment?

## 5.1 Basic Experiment Setting

An overview of the basic setup is provided. See Appendix B for details.

**Tasks and Environments.** We utilize established safety reinforcement learning tasks within the **BulletSafetyGym** environment following [30]. In our experiment, we focus on two tasks: Circle and Run, and train three types of agents: Car, Ant, and Drone.

**Dataset.** We utilize the offline safe RL dataset from [29]. For each environment, the dataset is collected by training different implemented algorithms under gradually varied cost thresholds and collecting the generated trajectories. The algorithms employed during this procedure include CPO [1], FOCOPS [45], CVPO [28], among others.

**Baselines and Parameters Setting.** In our selection of baseline models, we have covered a variety of established offline safe RL methods, as thoroughly discussed in [29]. The baselines encompass **CDT** [30], **Behavior Cloning (BC)** [41], **COptiDICE** [24], **CPQ** [41], **BCQ-Lag** [11, 37], and **BEAR-Lag**[21, 37].

## 5.2 CQDT Performance Analysis

**Evaluation Metrics.** We employ the normalized reward return and normalized cost return as our comparison metrics [10]. Let $R_{\max}(\mathcal{T})$ and $R_{\min}(\mathcal{T})$ denote the maximum and minimum empirical reward returns for task $\mathcal{M}$, respectively. The normalized reward is calculated as $R_{\text{norm}}(\mathcal{T}) = \frac{R_{\pi}(\mathcal{T}) - R_{\min}(\mathcal{T})}{R_{\max}(\mathcal{T}) - R_{\min}(\mathcal{T})}$, where $R_{\pi}$ represents the raw return of policy $\pi$. For the cost measure, we use the normalized cost return defined as $C_{\text{norm}}(\mathcal{T}) = \frac{C_{\pi}}{\kappa + \epsilon}$, where $\kappa$ denotes the cost limitation of the task and $\epsilon$ is a small positive number incorporated to ensure numerical stability. The term $C_{\pi}$ represents the evaluated accumulated cost return of policy $\pi$. If the cost return surpasses $\kappa + \epsilon$, the normalized cost return exceeds one; otherwise, it remains within the range of one.

**Result Analysis.** Table 1 provides a comprehensive overview of the performance of contemporary offline RL strategies in various environments and task configurations. Performance is assessed using reward and cost metrics, with evaluation criteria outlined as follows:

- When $C_{\text{norm}} \leq 1$, a larger $R_{\text{norm}}$ indicates superior strategy performance.

- When $C_{\text{norm}} \geq 1$, a smaller $C_{\text{norm}}$ signifies enhanced strategy performance.

- Comparing strategies with $C_{\text{norm}} \leq 1$ and $C_{\text{norm}} \geq 1$, preference is given to the policy with $C_{\text{norm}} \leq 1$.

Compared to other baselines, our proposed CQDT method achieves maximum return while ensuring safety. CPQ, BCQ-Lag, and BEAR-Lag, three Q-learning-based safe reinforcement learning methods, encounter challenges in balancing safety and reward optimization. The BC-Safe method, grounded in imitation learning and trained on provided data, exhibits suboptimal performance during the test phase. This phenomenon may be attributed to the scarcity of safe data in the training dataset, indicating a lack of robustness. COptiDICE employs novel estimators to evaluate policy constraints and achieves suboptimal rewards, while facing challenges related to adhering to strict safety constraints.

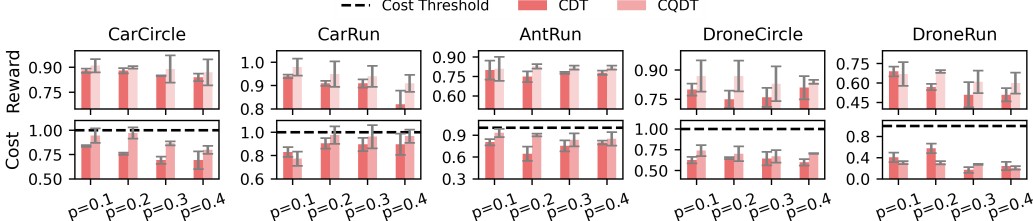

**Figure 5:** Verification of stitching-reward ability with $p = 0.x$ values representing various suboptimal datasets. Suboptimal datasets were generated by removing safe trajectories that fall within the top x% of cumulative rewards. Higher $p$-values indicate the removal of more high-reward safe paths, degrading dataset quality. The first row illustrates cumulative rewards obtained by CQDT and CDT trained on these datasets for different tasks, while the second row shows the corresponding cumulative costs. The black dashed line denotes the cost limitation.

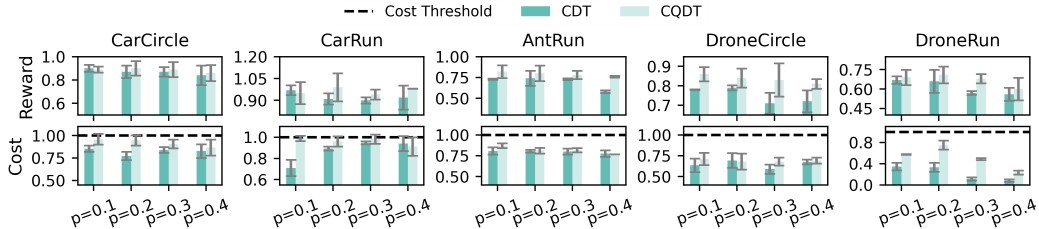

**Figure 6:** Verification of stitching-cost ability with $p = 0.x$ values corresponding to various suboptimal datasets. These datasets were created by excluding trajectories with low cumulative costs. As the $p$-value increases, fewer trajectories with small cumulative costs are retained, resulting in increasingly suboptimal datasets.

The CQDT method presented in our work leverages additional value functions to relabel trajectories, enhancing the model's stitching capability and enabling it to achieve state-of-the-art performance. To demonstrate the effectiveness of the CQDT method, we conduct a comprehensive comparative analysis examining the impact of various value functions on algorithmic performance. BCQ-Lag and BEAR-Lag, both grounded in offline safe reinforcement learning and based on Q-Learning, are employed for this investigation. The $Q_r$ and $Q_c$ functions in these methodologies are employed to estimate the RTG and CTG values of the original trajectory. To signify the enhanced versions of these methods, we specifically refer to them as **BCQL-CDT** and **BEARL-CDT**, respectively. Refer Appendix C.2 for further details.

Our experimental results, detailed in Table 1, indicate that BCQL-CDT and BEARL-CDT perform similarly to CDT in the selected tasks, although they do not reach the performance of CQDT. The performance discrepancy between BCQ-Lag and BEAR-Lag compared to CPQ suggests suboptimality in relabeling the original trajectories using their respective value functions. This contributes to the varied performance among BCQL-CDT, BEARL-CDT, and CQDT.

We also conduct two ablation studies, Ablation ① and Ablation ②, to assess the impact of various components within CQDT. The results of these experiments are provided in Table 1. For further details on the ablation studies, please refer to Appendix C.1. In addition, we evaluate the Zero-Shot Adaptation capability and robustness of CQDT. For more information, refer to Appendix C.4.

## 5.3 The Stitching Capability of CQDT

We conduct an evaluation of CQDT's stitching capabilities for reward and cost by creating various suboptimal datasets for five tasks and comparing the performance of CQDT and CDT across these datasets.

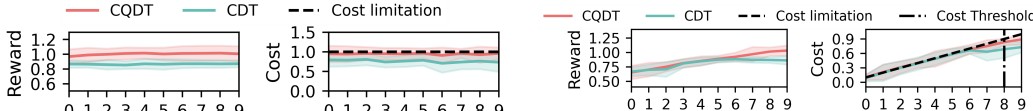

**Figure 7:** Comparison of CQDT and CDT performance in the Sparse-Reward DroneCircle environment with varying target return and fixed target cost.

**Figure 8:** Comparison of CQDT and CDT performance in the Sparse-Reward DroneCircle environment with varying target costs and fixed target return.

To evaluate the reward stitching ability, we remove the top X% of trajectories with the highest RTG from batches of trajectories. As the value of X increases, more high-return trajectories are excluded from the dataset. To create a suboptimal dataset for evaluating cost stitching capability, we group trajectories based on their RTG, then we remove the trajectories with lowest CTG in each group. In detail, we divide the trajectories into $\frac{Max\ Return}{10}$ groups, where Max Return denotes the highest return among all trajectories. Within each group, we remove trajectories with the lowest X% CTG from each group. Such a setup allows us to detect the stitching ability for a safe offline RL agent, as we want her to learn safe and high-return policy from unsafe and low-return trajectories. We leave the detailed parameter setup and the visualization of these suboptimal datasets in the Appendix C.3.

We present the performance of CQDT on different suboptimal datasets in Figures 5 and 6. These experiments demonstrate that as the value of $X$ increases, leading to the removal of higher-quality trajectories, the performance of policies generated by CQDT and CDT deteriorates. We can observe that the cumulative reward decreases. However, CQDT consistently outperforms CDT, demonstrating its superior stitching ability. Even when trained with suboptimal datasets, CQDT effectively utilizes these datasets to maximize performance by leveraging its stitching capabilities. Superior performance by CQDT highlights its unique ability to stitch suboptimal trajectories, a capability not present in CDT. This stitching ability enables CQDT to achieve better overall performance.

### 5.4 Performance of CQDT in Sparse Reward Environment

The experiments in the previous sections demonstrate that CQDT performs well in dense reward environments. In this section, we evaluate and analyze the performance of CQDT in the sparse reward environment. Since there is no publicly available dataset or corresponding environment for sparse reward scenarios in the field of offline safe RL, we build our own environment based on existing datasets. Specifically, we select the existing DroneCircle environment to create a Sparse-Reward DroneCircle environment.

For any trajectory $\tau$ in DroneCircle, we aim to create a new trajectory $\tau'$ as follows. We consider each subsequence in $\tau$ with length 10, i.e., $\tau.r_{10k}, \tau.r_{10k+1}, ..., \tau.r_{10k+9}$. We then set $\tau' = \tau$, while it replaces $\tau'.r_{10k+9}$ with $\tau.r_{10k} + \tau.r_{10k+1} + \cdots + \tau.r_{10k+9}$, and replaces $\tau'.r_{10k+i}, 0 \leq i < 9$ with 0. Such an operation keeps the total reward of $\tau'$ unchanged, while it makes $\tau'$ a trajectory with sparse rewards. Accordingly, we use the Sparse-Reward DroneCircle Offline Dataset to train CQDT and CDT. During testing, we adopt the similar strategy, where each agent encounters 0 reward in time step $10k, 10k + 1, ..., 10k + 8$, and she encounters $\tau.r_{10k} + \tau.r_{10k+1} + \cdots + \tau.r_{10k+9}$ at time step $10k + 9$. We do not change the cost distribution.

Figures 7 and 8 show the performance comparison between CQDT and CDT under different target return and target cost settings in the Sparse-Reward DroneCircle environment. The results indicate that CQDT consistently outperforms CDT across various settings of target return and target cost. These experimental results demonstrate that CQDT maintains its superiority even in sparse-reward environments.

## 6 Conclusion and Future Work

In this work, we proposed the Constrained Q-learning Decision Transformer (CQDT) for safe offline RL. Our approach replaces reward-to-go and cost-to-go in the training data with dynamic programming-based learning-based reward return and cost return, which brings the stitching ability and addresses the weakness of the Constrained Decision Transformer (CDT). Our evaluation shows that our approach is able to outperform existing safe RL baseline algorithms. One potential future direction is to build a theoretical analysis to justify the effectiveness of our learning-based constraint approach to safe RL, similar to previous analyses applied to general goal-based RL algorithms [5].

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

# A  Implementation Details

In this section, we introduce more details of the implementation of CQDT.

## A.1  Implementation Details of CPQ (First Step of CQDT)

**OOD discriminator $\nu$.** The process of learning the OOD action generation distribution $\nu$ uses an OOD detection method based on the Conditional Variational Autoencoder (CVAE). In detail, $\nu$ is based on a decoder $p : \mathcal{S} \times \mathcal{Z} \times \mathcal{A} \to [0, 1]$ that generates the action following the distribution $p(s, z, \cdot)$, where $z \in \mathcal{Z} \subseteq \mathbb{R}$ is the hidden state; an encoder $q : \mathcal{S} \times \mathcal{A} \times \mathcal{Z} \to [0, 1]$ that generates the latent state following the distribution $q(s, a, \cdot)$. $p$ and $q$ are trained by solving the following evidence lower bound (ELBO) objective, which is

$$\max_{p,q} \mathbb{E}_{s,a\sim\mathcal{T}} \Big[ \mathbb{E}_{z\sim q} \log p(s, z, a) - \beta \mathrm{KL}(q(s, a, \cdot) \| N(0, 1)) \Big], \tag{4}$$

where KL denotes the KL divergence and $\beta$ is the penalty parameter. We then set $\nu$ as

$$\nu(s, a) = \begin{cases} 1 & \mathrm{KL}(q(s, a, \cdot) \| N(0, 1)) \geq d \\ 0 & \text{Otherwise} \end{cases} \tag{5}$$

Here we provide a detailed implementation code for CPQ, following **(author?)** [41].

---

**Algorithm 3** CPQ

---

**Require:** Trajectories dataset $\mathcal{T}$; constraint limitation $\kappa$; initialize encoder $q$ and decoder $p$; training steps $M$ and $N$.
1: // VAE training
2: **for** $t = 0$ to $M$ **do**
3:    Sample mini-batch of state-action pairs $(s, a) \sim \mathcal{T}$ and update $p, q$ through (4)
4: **end for**
5: Update $\nu$ following (5)
6: // Policy training
7: Initialize reward critic $Q_r$, cost critic $Q_c$, actor $\pi_\theta$
8: **for** $t = 0$ to $N$ **do**
9:    Update cost critic by

$$Q_c = \arg\min_Q -\alpha \mathbb{E}_{s,a\sim\mathcal{T}}[Q(s, a)\nu(s, a)] + \mathbb{E}_{s,a,s',c\sim\mathcal{T}}[(Q(s, a) - c - \gamma\mathbb{E}_{a'\sim\pi(\cdot|s')}[Q(s', a')])^2],$$

10:    Update reward critic by

$$Q_r = \arg\min_Q \mathbb{E}_{s,a,s',r\sim\mathcal{T}}[(Q(s, a) - r - \gamma\mathbb{E}_{a'\sim\pi}[\mathbb{I}(Q_c(s', a') \leq \kappa)Q(s', a')])^2],$$

11:    Update policy as

$$\pi' = \arg\max_\pi \mathbb{E}_{s\sim\mathcal{T}}\mathbb{E}_{a\sim\pi(\cdot|s)}[\mathbb{I}(Q_c(s, a) \leq \kappa)Q_r(s, a)].$$

12: **end for**
**Ensure:** reward critic $Q_r(s, a)$, cost critic $Q_c(s, a)$

---

## A.2  Implementation details of Data Augmentation (Third Step of CQDT)

We adopt a data augmentation technique based on the Pareto Frontier (PF) from [30]. The dataset used in safe offline RL is characterized by the PF value $\mathrm{PF}(\kappa, \mathcal{T})$, which is

$$\mathrm{PF}(\kappa, \mathcal{T}) = \max_{\tau \in \mathcal{T}} R(\tau), \quad \text{s.t. } C(\tau) \leq \kappa$$

Given that CQDT shares the fundamental framework with CDT, it adopts the target returns-conditioned policy structure. Consequently, the agent's behavior becomes sensitive to choices regarding target return and cost. This sensitivity limits the valid choices for target cost and reward

**Table 2:** The hyperparameter configurations for dataset collection algorithms across diverse tasks. The parameters ***Cost Start***, ***Cost End***, ***Epoch Start***, and ***Epoch End*** are utilized to modify the cost limitations.

| Task | Cost Start | Cost End | Epoch Start | Epoch End | Epoch Number | Max Trajectory Length |
|------|-----------|----------|-------------|-----------|--------------|----------------------|
| CarCircle | 5 | 100 | 100 | 900 | 1000 | 1500 |
| CarRun | 5 | 100 | 100 | 900 | 1000 | 1500 |
| AntRun | 5 | 200 | 400 | 2500 | 2600 | 2000 |
| DroneCircle | 5 | 150 | 500 | 2500 | 2600 | 2000 |
| DroneRun | 100 | 5 | 50 | 800 | 1000 | 1500 |

return pairs to the PF points. To address this limitation, CQDT employs the same data enhancement strategy as CDT to improve the relabeled data. We list the details of the data augmentation technique in Algorithm 4.

---

**Algorithm 4** PF augmentation

---

**Require:** Trajectories dataset $\mathcal{T}$, iteration number $N$, max length $T$ of trajectories in $\mathcal{T}$
1: Set $c_{\min} = \min_{\tau \in \mathcal{T}} C(\tau)$, $c_{\max} = \max_{\tau \in \mathcal{T}} C(\tau)$, $r_{\max} = \max_{\tau \in \mathcal{T}} R(\tau)$
2: **for** $i = 1, \ldots, N$ **do**
3:    $\kappa_i \sim \text{Uniform}(c_{min}, c_{max})$ // Sample a cost return
4:    $\rho_i \sim \text{Uniform}(\text{PF}(\kappa_i, \mathcal{T}), r_{max})$ // Sample a reward return above the PF value
5:    $\tau_i^* \leftarrow \arg\max_{\tau \in \mathcal{T}} R(\tau)$,    s.t. $C(\tau) \leq \kappa_i$ // Find the closest and safe Pareto trajectory
6:    Generate $\hat{\tau}_i$, where $\hat{\tau}_i.R_t \leftarrow \tau_i^*.R_t + \rho_i - R(\tau_i^*)$, $\hat{\tau}_i.C_t \leftarrow \tau_i^*.C_t + \kappa_i - C(\tau_i^*)$ for all $1 \leq t \leq T$ // Relabel the reward and cost return
7:    $\mathcal{T} \leftarrow \mathcal{T} \cup \{\hat{\tau}_i\}$ // Append the trajectory to the dataset
8: **end for**
**Ensure:** Augmented trajectories dataset $\mathcal{T}$

---

Finally, we generate the final sliced trajectories dataset $\mathcal{T}^K$ which includes consistent trajectory $\tau$ with length $K$. To generate it, for each $\tau \in \mathcal{T}$, we regenerate new trajectories dataset $\tau^1, \ldots, \tau^{T-K}$ as follows. For $\tau^t$, we set its last RTG and CTG as $\tau^t.R_{t+K} \leftarrow \tau.R_{t+K}$ and $\tau^t.C_{t+K} \leftarrow \tau.C_{t+K}$. Then we repeatedly apply $\tau^t.R_i \leftarrow \tau.r_i + \tau^t.R_{i+1}$ and $\tau^t.C_i \leftarrow \tau.c_i + \tau^t.C_{i+1}$ for $i = t + K - 1, \ldots, t$. We summarize it in Algorithm 5.

# B  Experiment Setting and Hyperparameters

## B.1  Tasks Description

In our experiments, we employ the BulletSafetyGym environment, which is a suite built atop the PyBullet physics simulator, resembles *SafetyGym* but features shorter horizons and a larger number of agents [1, 34, 9, 8, 6]. Within BulletSafetyGym, we deploy three distinct agent models: the Car, Ant, and Drone. The Ant agent is designed as a four-legged creature with a spherical torso, while the Car agent, inspired by MIT's Racecar, features a four-wheeled configuration. The Drone agent is an aerial vehicle based on the AscTec Hummingbird quadrotor. These agents are employed to complete the Circle and Run tasks. In the Circle task, they move clockwise on a circle. The reward is defined as

$$r(s) = \frac{v^T[-y, x]}{1 + 3|r_{agent} - r_{circle}|} \tag{6}$$

(6) is maximized when agents move quickly in a clockwise direction. Costs are incurred if the agent leaves the safety zone defined by the two yellow boundaries, i.e., $c(s) = \mathbb{I}[|x| > x_{lim}]$, where $x$ is the position on the x-axis. In the Run task, agents earn rewards for navigating an avenue located between two non-physical, penetrable safety boundaries that incur costs upon breach. Additional costs are applied if agents surpass an agent-specific velocity threshold.

## B.2  Dataset Collection

The dataset collection process is the same as [29], and we include it for completeness. In the data collection process, we use various algorithms and distinct cost thresholds tailored to each envi-

**Algorithm 5** Consistent RTG and CTG

---

**Require:** Trajectories dataset $\mathcal{T}$, context length $K$
1: Set $\mathcal{T}^K = \emptyset$
2: **for** $\tau \in \mathcal{T}$ **do**
3:     **for** $t = 1, \ldots, T - K$ **do**
4:         Set $\tau^t = \{\tau.s_t, \tau.a_t, \tau.r_t, \tau.c_t, \ldots, \tau.s_{t+K}, \tau.a_{t+K}, \tau.r_{t+K}, \tau.c_{t+K}\}$
5:         Set $\tau^t.R_{t+K} \leftarrow \tau.R_{t+K}$ and $\tau^t.C_{t+K} \leftarrow \tau.C_{t+K}$
6:         **for** $i = t + K - 1, \ldots, t$ **do**
7:             Set $\tau^t.R_i \leftarrow \tau.r_i + \tau^t.R_{i+1}$ and $\tau^t.C_i \leftarrow \tau.c_i + \tau^t.C_{i+1}$
8:         **end for**
9:         $\mathcal{T}^K \leftarrow \mathcal{T}^K \cup \{\tau^t\}$
10:    **end for**
11: **end for**
**Ensure:** Sliced trajectories dataset $\mathcal{T}^K$

---

**Table 3:** Description of the experimental datasets: **_Max TS_** denotes the maximum length of an episode, while **_Act. Space_** and **_State Space_** represent the dimensions of action and state, respectively. **_MaxCost_**, **_MinCost_**, **_MaxReward_**, and **_MinReward_** represent the cumulative reward and cumulative cost obtained by each trajectory. **_Traj._** indicates the number of trajectories in the dataset.

| Bench. | Task | Max TS | Act. Space | State Space | MaxCost | MinCost | MaxReward | MinReward | Traj. |
|---|---|---|---|---|---|---|---|---|---|
| | CarCircle | 300 | 2 | 8 | 100 | 0 | 534.306 | 3.484 | 1450 |
| | CarRun | 200 | 2 | 7 | 40 | 0 | 574.653 | 204.287 | 651 |
| BulletSafetyGym | AntRun | 200 | 8 | 33 | 150 | 0 | 955.481 | 0 | 1816 |
| | DroneCircle | 300 | 4 | 18 | 100 | 0 | 996.389 | 207.79 | 1923 |
| | DroneRun | 200 | 4 | 17 | 140 | 0 | 682.83 | 10.557 | 1990 |

ronment. The algorithms utilized for dataset acquisition include CPO, FOCOPS, PPOLagrangian, TRPOLagrangian, DDPGLagrangian, SACLagrangian, and CVPO. Notably, PPOLagrangian, TR-POLagrangian, DDPGLagrangian, and SACLagrangian are composite methods combining PPO [1], TRPO [35], DDPG [26], SAC [15], and PID Lagrangian [37] techniques, respectively. Among these algorithms, the first four belong to the category of On-Policy algorithms, while DDPGLagrangian and SACLagrangian fall under Off-On-Policy algorithms. On the other hand, CVPO is classified as an Off-Policy algorithm. Table 2 outlines the hyperparameters utilized for collecting datasets across different tasks, and Table 3 presents detailed information about the constructed datasets.

## B.3 Hyperparameters

We list the hyperparameters for CQDT here as well as the hyperparameters employed in the training of CPQ. Specifically, within CPQ, we utilize the $Q_r$ and $Q_c$ functions for trajectory relabeling. The hyperparameters for CPQ training are listed in Table 4. Meanwhile, the hyperparameters for CDT and CQDT training are comprehensively detailed in Table 5.

# C  Result Details and Discussions

## C.1  Ablation Study

In assessing the impact of various components within CQDT, we conducted two ablation studies.

**Joint Relabeling.** This is denoted as Ablation ①, aiming to study whether our adopted relabeling strategy in Algorithm 2 is necessary. We compare it with an alternative relabeling approach which does not take both the reward and cost into consideration together. Instead, the alternative relabeling approach relabels the RTG and CTG similar to what QDT does, which relabels them separately. For a trajectory $\tau$, the alternative approach obtains $V_r^\tau$ and $V_c^\tau$ the same as Algorithm 2, then for each $t = T + 1, \ldots, 2$, it relabels $\tau.R_{t-1} \rightarrow \tau.r_{t-1} + \max(\tau.R_t, V_r^\tau(\tau.s_t))$ when $V_r^\tau(s_{\mathcal{L}}) \geq \tau.R_{\mathcal{L}}$, and it relabels $\tau.C_{t-1} \rightarrow \tau.c_{t-1} + \min(\tau.C_t, V_c^\tau(\tau.s_t))$ when $V_c^\tau(s_t) \leq \tau.C_t$. The main difference between such a relabeling strategy and our adopted strategy for CQDT is the separate consideration for reward and cost relabeling. Our result reveals that with such a relabeling process, CQDT performance is notably worse in scenarios such as AntRun and DroneCircle, which exhibit poor CPQ performance

**Table 4:** Parameters utilized in training CPQ. $range(x : y, z)$ means the cost limitation begins at $x$ and increases by $z$ each step until it approaches $y$.

| | CarCircle | CarRun | AntRun | DroneCircle | DroneRun |
|---|---|---|---|---|---|
| $state\_dim$ | 8 | 7 | 33 | 18 | 17 |
| $action\_dim$ | 2 | 2 | 8 | 4 | 4 |
| $max\_action$ | 1.0 | 1.0 | 1.0 | 1.0 | 1.0 |
| $actor\_hidden\_size$ | $[256, 256]$ | $[256, 256]$ | $[256, 256]$ | $[256, 256]$ | $[256, 256]$ |
| $critic\_hidden\_size$ | $[256, 256]$ | $[256, 256]$ | $[256, 256]$ | $[256, 256]$ | $[256, 256]$ |
| $VAE\_hidden\_size$ | 400 | 400 | 400 | 400 | 400 |
| $sample\_action\_num$ | 10 | 10 | 10 | 10 | 10 |
| $Q_r\ number$ | 2 | 2 | 2 | 2 | 2 |
| $Q_c\ number$ | 2 | 2 | 2 | 2 | 2 |
| $episode\_len$ | 300 | 200 | 200 | 300 | 200 |
| $batchsize$ | 2048 | 2048 | 2048 | 2048 | 2048 |
| $update\_steps$ | 100000 | 100000 | 100000 | 100000 | 100000 |
| $vae\_lr$ | 0.001 | 0.001 | 0.001 | 0.001 | 0.001 |
| $critic\_lr$ | 0.001 | 0.001 | 0.001 | 0.001 | 0.001 |
| $actor\_lr$ | 0.0001 | 0.0001 | 0.0001 | 0.0001 | 0.0001 |
| $cost\_limitation$ | $(10 : 100, 10)$ | $(5 : 45, 5)$ | $(15 : 150, 15)$ | $(10 : 110, 10)$ | $(15 : 135, 15)$ |

**Table 5:** Parameters utilized in training CDT and CQDT. Shared parameters are denoted in black fonts, while the independent parameters of CQDT are marked in blue fonts.

| | CarCircle | CarRun | AntRun | DroneCircle | DroneRun |
|---|---|---|---|---|---|
| $state\_dim$ | 8 | 7 | 33 | 18 | 17 |
| $action\_dim$ | 2 | 2 | 8 | 4 | 4 |
| $max\_action$ | 1.0 | 1.0 | 1.0 | 1.0 | 1.0 |
| $embedding\_dim$ | 128 | 128 | 128 | 128 | 128 |
| $seq\_len$ | 10 | 10 | 10 | 10 | 10 |
| $episode\_len$ | 300 | 200 | 200 | 300 | 200 |
| $num\_layers$ | 3 | 3 | 3 | 3 | 3 |
| $num\_heads$ | 8 | 8 | 8 | 8 | 8 |
| $attention\_dropout$ | 0.1 | 0.1 | 0.1 | 0.1 | 0.1 |
| $residual\_dropout$ | 0.1 | 0.1 | 0.1 | 0.1 | 0.1 |
| $embedding\_dropout$ | 0.1 | 0.1 | 0.1 | 0.1 | 0.1 |
| $action\_head\_layers$ | 1 | 1 | 1 | 1 | 1 |
| $target\&cost\_return$ | $[500, (0 : 120, 10)]$ | $[575, (0 : 50, 5)]$ | $[900, (0 : 200, 20)]$ | $[900, (20 : 120, 20)]$ | $[600, (0 : 200, 20)]$ |
| $target\&cost\_return$ | $[(0 : 600, 50), 90]$ | $[(70 : 700, 70), 35]$ | $[(0 : 1100, 100), 130]$ | $[(300 : 1400, 100), 90]$ | $[(0 : 1000, 100), 120]$ |
| $batchsize$ | 2048 | 2048 | 2048 | 2048 | 2048 |
| $learning\_rate$ | 0.0001 | 0.0001 | 0.0001 | 0.0001 | 0.0001 |
| $update\_steps$ | 100000 | 100000 | 100000 | 100000 | 100000 |
| $weight\_decay$ | 0.0001 | 0.0001 | 0.0001 | 0.0001 | 0.0001 |

when cost return and reward return are considered independently. This indicates that the accuracy of value function predictions significantly influences model performance during the relabeling process, while our proposed CQDT method effectively minimizes the negative impacts caused by inaccuracies in value function predictions.

**PF Augmentation.** The second study, denoted as Ablation ②, evaluates CQDT without the PF-based augmentation step [30]. Our results reveal that the PF-based augmentation is essential for ensuring the model's security under various target cost settings. For instance, in the CarRun task, the absence of the PF augmentation technique results in a policy that exceeds cost limitations.

## C.2 Detailed Results from Value Function Experiment

In this section, we explore the utilization of diverse value functions for state value estimation, substituting RTG and CTG in the original trajectories. We then retrain our CQDT model with the replaced dataset and analyze the differences in results. We systematically investigate the incorporation of $Q_r$ and $Q_c$ as value functions for state value estimation in CPQ, BCQ-Lagrangian (BCQL), and BEAR-Lagrangian (BEARL). The algorithmic details of CPQ are outlined in Algorithm 3. Combining BCQ and BEAR with the Lagrangian approach, which utilizes adaptive penalty coefficients to enforce constraints, results in the formulation of BCQ-Lagrangian and BEAR-Lagrangian.

**Implementation Details of BCQ** For details regarding the BCQ, please refer to Algorithm 6. BCQ requires a tuple dataset $B = \{(s, a, r, s')\}$ and returns policy $\pi$ and reward critic $Q$. Here, $\pi$ is

503   defined based on $n$ sampled action $a_i$, where

$$\pi(s) = \arg \max_{a_i + \xi_\phi(s, a_i, \Phi)} Q_{\theta_1}(s, a_i + \xi_\phi(s, a_i, \Phi)), \ \{a_i \sim G_w(s)\}_{i=1}^n. \tag{7}$$

---

**Algorithm 6** BCQ Training

---

**Require:** Dataset $B$, iteration number $M$, target network update rate $\tau$, mini-batch size $N$, max perturbation $\Phi$, number of sampled actions $n$, minimum weighting $\lambda$.
1: Initialize $Q$-networks $Q_{\theta_1}, Q_{\theta_2}$, perturbation network $\xi_\phi$, and VAE $G_\omega = \{E_{\omega_1}, D_{\omega_2}\}$, with random parameters $\theta_1, \theta_2, \phi, \omega$, and target networks $Q'_{\theta_1}, Q'_{\theta_2}, \xi'_\phi$ with $\theta'_1 \leftarrow \theta_1, \theta'_2 \leftarrow \theta_2, \phi' \leftarrow \phi$.
2: **for** $t = 1, \ldots, M$ **do**
3:     Sample mini-batch of $N$ transitions $(s, a, r, s')$ from $B$
4:     $\mu, \sigma = E_{\omega_1}(s, a), \tilde{a} = D_{\omega_2}(s, z), z \sim \mathcal{N}(\mu, \sigma)$
5:     $\omega \leftarrow \arg \min_\omega \sum (a - \tilde{a})^2 + \text{KL}(\mathcal{N}(\mu, \sigma) \parallel \mathcal{N}(0, 1))$
6:     Sample $n$ actions: $\{\tilde{a}_i\}_{i=1}^n \sim G_\omega(s')$
7:     Perturb each action: $\{\tilde{a}_i = a_i + \xi_\phi(s', a_i, \Phi)\}_{i=1}^n$
8:     Set value target $y = r + \max_i[\lambda \min_{j=1,2} Q'_{\theta'_j}(s', \tilde{a}_i) + (1 - \lambda) \max_{j=1,2} Q'_{\theta'_j}(s', \tilde{a}_i)]$
9:     $\theta \leftarrow \arg \min_\theta \sum (y - Q_\theta(s, a))^2$ and $\phi \leftarrow \arg \max_\phi \sum Q_{\theta_1}(s, a + \xi_\phi(s, a, \Phi))$
10:    Update target networks: $\theta'_i \leftarrow \tau\theta_i + (1 - \tau)\theta'_i, \phi' \leftarrow \tau\phi + (1 - \tau)\phi'$
11: **end for**
**Ensure:** Policy $\pi$ follows (7), reward critic $Q = Q_{\theta_1}$

---

504   **Implementation Details of BEAR** For BEAR algorithm, we put its details in Algorithm 7. BEAR
505   takes a tuple dataset $B = \{(s, a, r, s')\}$ as its input. It operates based on the maximum mean
506   discrepancy (MMD) distance as follows

$$\text{MMD}^2(\{x_1, \cdots, x_n\}, \{y_1, \cdots, y_m\}) = \frac{1}{n^2} \sum_{i,i'} k(x_i, x_{i'}) - \frac{2}{nm} \sum_{i,j} k(x_i, y_j) + \frac{1}{m^2} \sum_{j,j'} k(y_j, y_{j'}). \tag{8}$$

---

**Algorithm 7** BEAR Training

---

**Require:** Dataset $B$, target network update rate $\tau$, iteration number $M$, sampled actions for MMD $n$, ratio parameter $\lambda$
1: Initialize Q-ensemble $\{Q_{\theta_i}\}_{i=1}^K$, actor $\pi_\phi$, Lagrange multiplier $\alpha$, target networks $\{Q'_{\theta_i}\}_{i=1}^K$, and a target actor $\pi'_\phi$ with $\phi' \leftarrow \phi, \theta'_i \leftarrow \theta_i$
2: **for** $t = 1, \ldots, M$ **do**
3:     Sample mini-batch of transitions $(s, a, r, s') \sim B$
4:     Sample $p$ action samples, $\{a_i \sim \pi'_\phi(s')\}_i^p$
5:     Define $y(s, a) = \max_{a_i}[\min_{j=1,\ldots,K} Q'_{\theta_j}(s', a_i) + (1 - \lambda) \max_{j=1,\ldots,K} Q'_{\theta_j}(s', a_i)]$
6:     $\forall i, \theta_i \leftarrow \arg \min_{\theta_i}((Q_{\theta_i}(s, a) - (r + \gamma y(s, a)))^2)$
7:     Sample actions $\{a_i \sim \pi_\phi(s)\}_{i=1}^m$ and $\{a_j \sim B(s)\}_{j=1}^n$, where $B(s)$ represents the set of actions appearing the the dataset $B$
8:     Update $\phi, \alpha$ by minimizing

$$\pi_\phi := \max_{\pi \in \Delta_{|S|}} \mathbb{E}_{s \sim B} \mathbb{E}_{a \sim \pi(\cdot|s)} \left[ \min_{j=1,\ldots,K} Q_{\theta_j}(s, a) \right] \quad \text{s.t.} \quad \mathbb{E}_{s \sim B}[\text{MMD}(B(s), \pi(\cdot|s))] \leq \epsilon, \tag{9}$$

    where the MMD distance is defined as in (8)
9:     Update Target Networks: $\theta'_i \leftarrow \tau\theta_i + (1 - \tau)\theta'_i, \phi' \leftarrow \tau\phi + (1 - \tau)\phi'$
10: **end for**
**Ensure:** Policy $\pi_\phi$ and reward critics $\{Q_{\theta_j}\}$

---

## C.3 Detailed Analysis of Stitching Capability Verification

We conduct a comprehensive evaluation of the cost and reward stitching abilities of the CQDT model. We summarize the parameters for testing the stitching ability in Table 6. To rigorously assess the cost stitching capabilities, we generate a suboptimal dataset by excluding the top $X\%$ of trajectories with the lowest cost returns from a set of trajectories that exhibit similar reward returns. Specifically, we group trajectories based on their RTG, then remove the trajectories with the lowest CTG in each group. In detail, we divide the trajectories into $\frac{\text{Max Return}}{10}$ groups, where Max Return denotes the highest return among all trajectories. Within each group, we remove trajectories with the lowest $X\%$ CTG. For the assessment of reward stitching capabilities, we similarly remove the top $X\%$ of trajectories with the highest reward returns from the trajectories. Figures 9 and 10 present visualizations of the datasets used to validate the reward and cost stitching abilities.

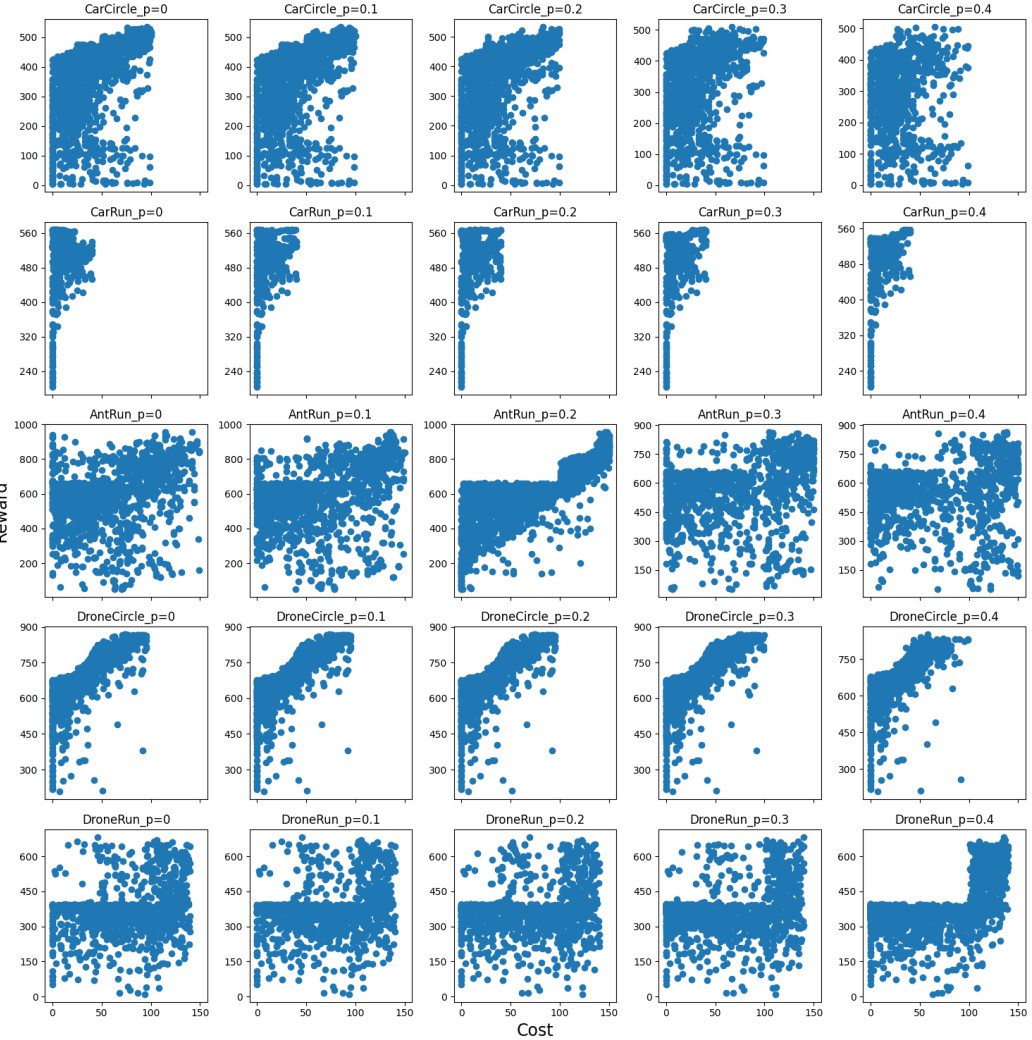

**Figure 9:** Visualization of the dataset used to validate reward stitching ability.

## C.4 Detailed Results from Zero-Shot Adaptation and Robustness Evaluation Experiment

We study the robustness and the zero-shot adaptation ability of CQDT in this section. Figure 11 depicts a fixed target cost, illustrating how variations in target return impact performance, providing a measure of robustness. Meanwhile, Figure 12 showcases a fixed target return, demonstrating how changes in target cost influence the actual performance, serving as an evaluation of zero-shot adaptation ability. For the raw experiment data, we include them in Table 8 and 9.

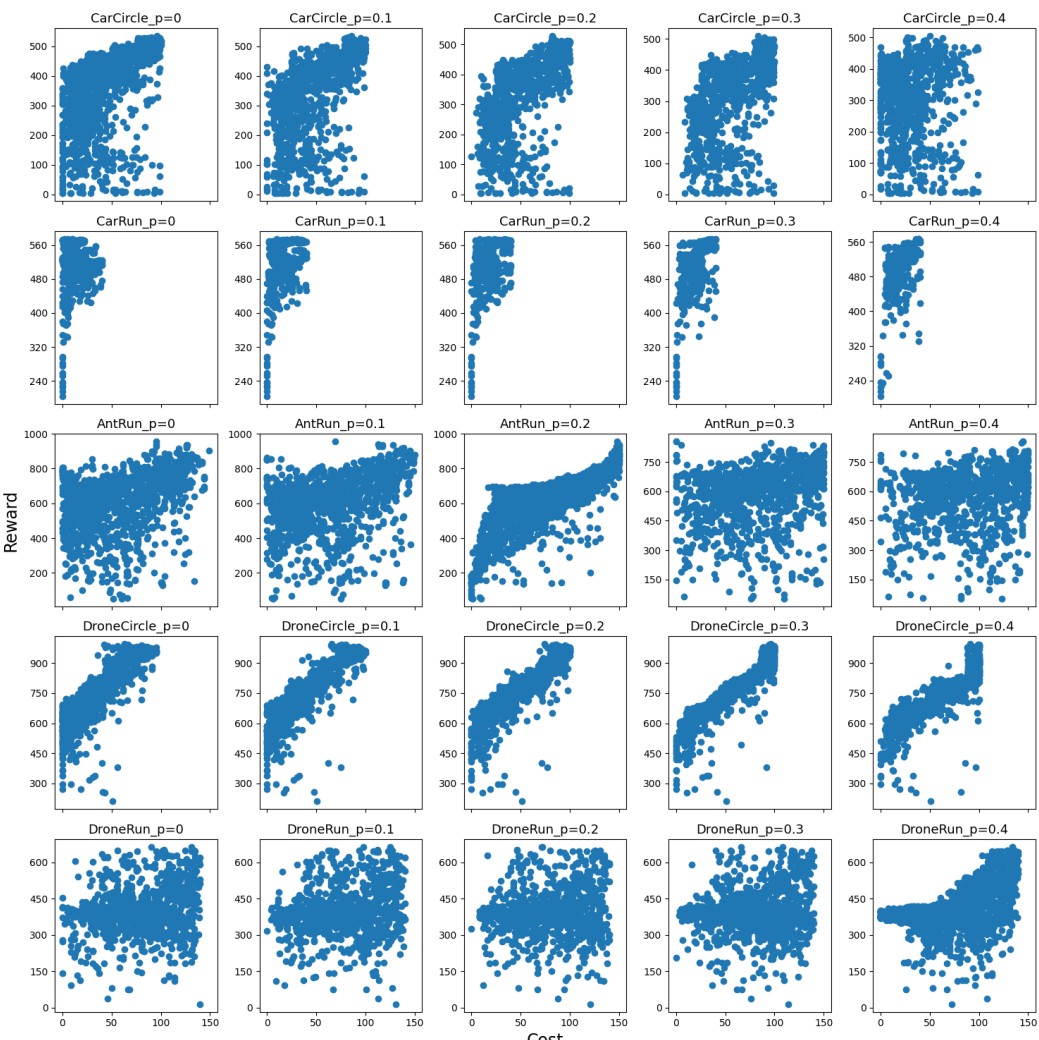

**Figure 10:** Visualization of the dataset used to validate cost stitching ability.

**Robustness Validation.** During the robustness validation phase, CQDT's performance initially benefits from maintaining a constant target cost while gradually increasing the target return, leading to an augmentation in cumulative reward without violating cost constraints. However, once the target return reaches a certain threshold, further increases do not result in a proportional rise in cumulative reward. This phenomenon occurs because the target return setting guides CQDT's prediction process but does not alter the intrinsic training process, which the model architecture and the training dataset determine. If the target return setting significantly exceeds the cumulative reward of the optimal trajectory in the training dataset, CQDT's performance may not experience substantial improvements. Despite CQDT's stitching ability, which allows cumulative rewards to increase when the target return setting exceeds the maximum threshold in the dataset, the positive effect of this ability is limited.

**Zero-shot Adaptation Validation.** In the zero-shot adaptation validation experiment, where the target return remains constant while the target cost varies, the analysis of cumulative rewards across five tasks reveals an initial increase followed by stabilization. As the target cost setting gradually increases, the constraints on cost become more relaxed, thereby enhancing the strategy's ability to maximize cumulative rewards during the learning process. However, when the target cost setting exceeds the maximum cumulative cost threshold in the dataset, further increases in the target cost do not enhance the strategy's ability to maximize cumulative rewards, leading to no further increase in cumulative rewards obtained during the test phase.

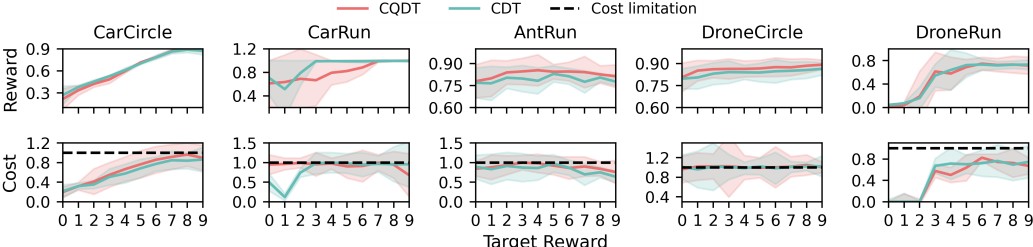

**Figure 11:** Results of robustness verification to different reward returns. Each column represents a task. The x-axis denotes the target return. The first row shows the evaluated reward, and the second row shows the evaluated cost, both under different target return. The dashed line represents the predefined cost limitation.

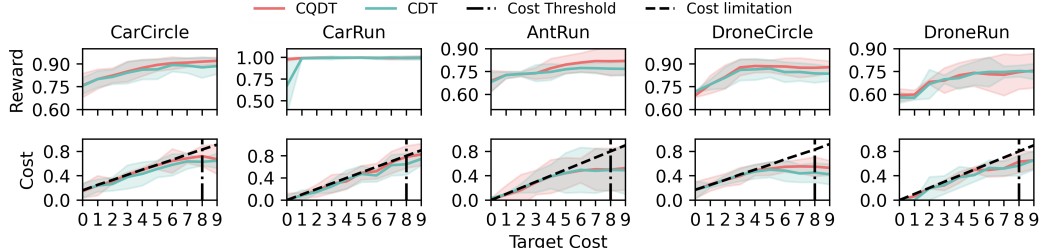

**Figure 12:** Results of zero-shot adaptation to different cost returns. Each column represents a task. The x-axis denotes the target cost return. The first row and the second row display the evaluated reward and cost under different target costs, respectively. The dashed line represents the predefined cost limitation, while the solid line indicates the maximum cost of trajectories included in the dataset.

**Table 6:** Parameter Settings for the Stitching Ability Validation Experiment

|           | CarCircle | CarRun | AntRun | DroneCircle | DroneRun |
|-----------|-----------|--------|--------|-------------|----------|
| $\kappa$        | 90        | 35     | 130    | 90          | 120      |
| Target Return | 500   | 575    | 900    | 900         | 600      |
| Target Cost   | 90    | 35     | 130    | 90          | 120      |

Indeed, while CQDT consistently outperforms CDT in both robustness and zero-shot adaptation experiments, the challenge of selecting appropriate target return and target cost values remains common to both approaches. Optimal choices for these parameters should align with the specific characteristics of the task training dataset. Setting a larger target return and a smaller target cost may not be advisable; a more viable approach involves tailoring these targets based on the inherent properties of the task-specific training data. This task-specific customization ensures a more effective and contextually appropriate utilization of the reinforcement learning framework.

**Table 7:** Settings of variables cX-r[01-10] and c[01-10]-rX in robustness validation and zero-shot adaptation experiments across different tasks. During the evaluation stage, the settings for Target Return and Target Cost correspond to the actual values in the environment, rather than the normalized values.

| Task | Experiment | 1 | 2 | 3 | 4 | 5 | 6 | 7 | 8 | 9 | 10 |
|------|-----------|---|---|---|---|---|---|---|---|---|----|
| CarCircle | Robustness | c90r100 | c90r150 | c90r200 | c90r250 | c90r300 | c90r350 | c90r400 | c90r450 | c90r500 | c90r550 |
|           | Zero-shot | c20r500 | c30r500 | c40r500 | c50r500 | c60r500 | c70r500 | c80r500 | c90r500 | c100r500 | c110r500 |
| CarRun | Robustness | c35r70 | c35r140 | c35r210 | c35r280 | c35r350 | c35r420 | c35r490 | c35r560 | c35r630 | c35r700 |
|        | Zero-shot | c0r575 | c5r575 | c10r575 | c15r575 | c20r575 | c25r575 | c30r575 | c35r575 | c40r575 | c45r575 |
| AntRun | Robustness | c130r0 | c130r100 | c130r200 | c130r300 | c130r400 | c130r500 | c130r600 | c130r700 | c130r800 | c130r900 |
|        | Zero-shot | c0r900 | c20r900 | c40r900 | c60r900 | c80r900 | c100r900 | c120r900 | c140r900 | c160r900 | c180r900 |
| DroneCircle | Robustness | c90r300 | c90r400 | c90r500 | c90r600 | c90r700 | c90r800 | c90r900 | c90r1000 | c90r1100 | c90r1200 |
|             | Zero-shot | c20r900 | c30r900 | c40r900 | c50r900 | c60r900 | c70r900 | c80r900 | c90r900 | c100r900 | c110r900 |
| DroneRun | Robustness | c120r0 | c120r100 | c120r200 | c120r300 | c120r400 | c120r500 | c120r600 | c120r700 | c120r800 | c120r900 |
|          | Zero-shot | c0r600 | c20r600 | c40r600 | c60r600 | c80r600 | c100r600 | c120r600 | c140r600 | c160r600 | c180r600 |

**Table 8:** Experimental data from the CQDT, encompassing both the Zero-shot adaptation experiment (c[01-10]-rx) and the Robustness validation experiment (cx-r[01-10]) across diverse tasks. In the Robustness validation experiment, the target return is systematically varied while maintaining a constant target cost. Conversely, in the Zero-shot adaptation experiment, the target return is held constant while adjusting the size of the target cost. Selection criteria for cx-r[01-10] and c[01-10]-rx in different tasks are detailed for each scenario, with specific task names and corresponding the above table.

| | CarCircle | | CarRun | | AntRun | | DroneCircle | | DroneRun | |
|---|---|---|---|---|---|---|---|---|---|---|
| | reward | cost | reward | cost | reward | cost | reward | cost | reward | cost |
| cX-r01 | $0.216_{\pm0.145}$ | $0.197_{\pm0.106}$ | $0.611_{\pm0.389}$ | $0.949_{\pm0.251}$ | $0.779_{\pm0.113}$ | $0.843_{\pm0.188}$ | $0.806_{\pm0.098}$ | $0.961_{\pm0.272}$ | $0.008_{\pm0.017}$ | $0.000_{\pm0.008}$ |
| cX-r02 | $0.335_{\pm0.068}$ | $0.316_{\pm0.079}$ | $0.634_{\pm0.463}$ | $0.964_{\pm0.150}$ | $0.799_{\pm0.124}$ | $0.878_{\pm0.314}$ | $0.852_{\pm0.070}$ | $1.020_{\pm0.280}$ | $0.040_{\pm0.053}$ | $0.000_{\pm0.150}$ |
| cX-r03 | $0.422_{\pm0.045}$ | $0.412_{\pm0.261}$ | $0.690_{\pm0.510}$ | $0.990_{\pm0.124}$ | $0.839_{\pm0.080}$ | $0.934_{\pm0.243}$ | $0.861_{\pm0.058}$ | $1.022_{\pm0.256}$ | $0.189_{\pm0.431}$ | $0.000_{\pm0.050}$ |
| cX-r04 | $0.484_{\pm0.055}$ | $0.549_{\pm0.252}$ | $0.666_{\pm0.436}$ | $0.956_{\pm0.301}$ | $0.846_{\pm0.090}$ | $0.999_{\pm0.185}$ | $0.863_{\pm0.069}$ | $1.025_{\pm0.631}$ | $0.615_{\pm0.328}$ | $0.576_{\pm0.124}$ |
| cX-r05 | $0.597_{\pm0.028}$ | $0.664_{\pm0.250}$ | $0.792_{\pm0.207}$ | $0.974_{\pm0.140}$ | $0.854_{\pm0.106}$ | $0.979_{\pm0.237}$ | $0.863_{\pm0.055}$ | $0.998_{\pm0.247}$ | $0.580_{\pm0.280}$ | $0.501_{\pm0.141}$ |
| cX-r06 | $0.715_{\pm0.005}$ | $0.764_{\pm0.279}$ | $0.822_{\pm0.184}$ | $0.901_{\pm0.299}$ | $0.844_{\pm0.050}$ | $0.922_{\pm0.232}$ | $0.866_{\pm0.056}$ | $0.994_{\pm0.206}$ | $0.690_{\pm0.130}$ | $0.637_{\pm0.246}$ |
| cX-r07 | $0.786_{\pm0.020}$ | $0.858_{\pm0.271}$ | $0.883_{\pm0.122}$ | $0.911_{\pm0.403}$ | $0.846_{\pm0.064}$ | $0.862_{\pm0.284}$ | $0.875_{\pm0.064}$ | $1.004_{\pm0.418}$ | $0.748_{\pm0.109}$ | $0.824_{\pm0.068}$ |
| cX-r08 | $0.868_{\pm0.022}$ | $0.915_{\pm0.255}$ | $0.991_{\pm0.020}$ | $0.997_{\pm0.203}$ | $0.841_{\pm0.107}$ | $0.899_{\pm0.247}$ | $0.874_{\pm0.057}$ | $1.017_{\pm0.205}$ | $0.725_{\pm0.114}$ | $0.744_{\pm0.148}$ |
| cX-r09 | $0.909_{\pm0.032}$ | $0.963_{\pm0.241}$ | $0.996_{\pm0.009}$ | $0.944_{\pm0.341}$ | $0.826_{\pm0.060}$ | $0.840_{\pm0.183}$ | $0.885_{\pm0.049}$ | $1.050_{\pm0.208}$ | $0.733_{\pm0.143}$ | $0.725_{\pm0.267}$ |
| cX-r10 | $0.900_{\pm0.043}$ | $0.896_{\pm0.257}$ | $0.997_{\pm0.010}$ | $0.676_{\pm0.553}$ | $0.814_{\pm0.077}$ | $0.758_{\pm0.296}$ | $0.892_{\pm0.033}$ | $1.003_{\pm0.108}$ | $0.716_{\pm0.146}$ | $0.672_{\pm0.153}$ |
| c01-rX | $0.759_{\pm0.079}$ | $0.148_{\pm0.111}$ | $0.978_{\pm0.019}$ | $0.000_{\pm0.120}$ | $0.689_{\pm0.069}$ | $0.015_{\pm0.020}$ | $0.693_{\pm0.012}$ | $0.163_{\pm0.138}$ | $0.596_{\pm0.040}$ | $0.002_{\pm0.013}$ |
| c02-rX | $0.800_{\pm0.064}$ | $0.237_{\pm0.163}$ | $0.993_{\pm0.007}$ | $0.088_{\pm0.052}$ | $0.727_{\pm0.024}$ | $0.085_{\pm0.155}$ | $0.765_{\pm0.059}$ | $0.258_{\pm0.108}$ | $0.599_{\pm0.029}$ | $0.063_{\pm0.022}$ |
| c03-rX | $0.824_{\pm0.049}$ | $0.309_{\pm0.041}$ | $0.997_{\pm0.005}$ | $0.168_{\pm0.112}$ | $0.733_{\pm0.014}$ | $0.164_{\pm0.255}$ | $0.814_{\pm0.078}$ | $0.329_{\pm0.104}$ | $0.682_{\pm0.017}$ | $0.199_{\pm0.011}$ |
| c04-rX | $0.850_{\pm0.067}$ | $0.412_{\pm0.047}$ | $0.997_{\pm0.002}$ | $0.293_{\pm0.167}$ | $0.739_{\pm0.039}$ | $0.289_{\pm0.200}$ | $0.877_{\pm0.054}$ | $0.395_{\pm0.138}$ | $0.691_{\pm0.082}$ | $0.241_{\pm0.084}$ |
| c05-rX | $0.875_{\pm0.054}$ | $0.498_{\pm0.035}$ | $0.996_{\pm0.010}$ | $0.338_{\pm0.142}$ | $0.772_{\pm0.063}$ | $0.387_{\pm0.275}$ | $0.886_{\pm0.043}$ | $0.493_{\pm0.091}$ | $0.726_{\pm0.082}$ | $0.374_{\pm0.092}$ |
| c06-rX | $0.894_{\pm0.040}$ | $0.573_{\pm0.069}$ | $0.998_{\pm0.006}$ | $0.494_{\pm0.126}$ | $0.796_{\pm0.058}$ | $0.442_{\pm0.255}$ | $0.885_{\pm0.040}$ | $0.516_{\pm0.109}$ | $0.742_{\pm0.088}$ | $0.496_{\pm0.049}$ |
| c07-rX | $0.905_{\pm0.041}$ | $0.644_{\pm0.198}$ | $0.994_{\pm0.013}$ | $0.520_{\pm0.140}$ | $0.809_{\pm0.062}$ | $0.485_{\pm0.200}$ | $0.883_{\pm0.055}$ | $0.547_{\pm0.087}$ | $0.732_{\pm0.109}$ | $0.495_{\pm0.116}$ |
| c08-rX | $0.909_{\pm0.029}$ | $0.686_{\pm0.080}$ | $0.992_{\pm0.019}$ | $0.681_{\pm0.199}$ | $0.819_{\pm0.077}$ | $0.506_{\pm0.345}$ | $0.875_{\pm0.064}$ | $0.553_{\pm0.114}$ | $0.728_{\pm0.136}$ | $0.523_{\pm0.133}$ |
| c09-rX | $0.915_{\pm0.025}$ | $0.723_{\pm0.169}$ | $0.996_{\pm0.020}$ | $0.780_{\pm0.200}$ | $0.818_{\pm0.096}$ | $0.500_{\pm0.350}$ | $0.875_{\pm0.054}$ | $0.549_{\pm0.159}$ | $0.746_{\pm0.113}$ | $0.631_{\pm0.183}$ |
| c10-rX | $0.920_{\pm0.023}$ | $0.672_{\pm0.245}$ | $0.998_{\pm0.018}$ | $0.824_{\pm0.196}$ | $0.820_{\pm0.091}$ | $0.526_{\pm0.360}$ | $0.879_{\pm0.040}$ | $0.530_{\pm0.120}$ | $0.756_{\pm0.114}$ | $0.653_{\pm0.140}$ |

**Table 9:** Experimental data from the CDT, encompassing both the Zero-shot adaptation experiment (c[01-10]-rx) and the Robustness validation experiment (cx-r[01-10]) across diverse tasks. Other settings are the same as CQDT.

| | CarCircle | | CarRun | | AntRun | | DroneCircle | | DroneRun | |
|---|---|---|---|---|---|---|---|---|---|---|
| | reward | cost | reward | cost | reward | cost | reward | cost | reward | cost |
| cX-r01 | $0.269_{\pm0.127}$ | $0.214_{\pm0.134}$ | $0.704_{\pm0.299}$ | $0.494_{\pm0.220}$ | $0.769_{\pm0.102}$ | $0.889_{\pm0.303}$ | $0.797_{\pm0.069}$ | $1.011_{\pm0.245}$ | $0.047_{\pm0.002}$ | $0.000_{\pm0.042}$ |
| cX-r02 | $0.379_{\pm0.022}$ | $0.318_{\pm0.061}$ | $0.514_{\pm0.489}$ | $0.119_{\pm0.093}$ | $0.765_{\pm0.116}$ | $0.830_{\pm0.385}$ | $0.805_{\pm0.079}$ | $0.959_{\pm0.374}$ | $0.071_{\pm0.016}$ | $0.000_{\pm0.158}$ |
| cX-r03 | $0.453_{\pm0.026}$ | $0.352_{\pm0.086}$ | $0.788_{\pm0.214}$ | $0.741_{\pm0.116}$ | $0.803_{\pm0.073}$ | $0.913_{\pm0.349}$ | $0.831_{\pm0.075}$ | $1.016_{\pm0.484}$ | $0.162_{\pm0.189}$ | $0.000_{\pm0.017}$ |
| cX-r04 | $0.528_{\pm0.009}$ | $0.507_{\pm0.166}$ | $0.990_{\pm0.014}$ | $0.983_{\pm0.274}$ | $0.798_{\pm0.089}$ | $0.872_{\pm0.359}$ | $0.841_{\pm0.077}$ | $1.009_{\pm0.258}$ | $0.546_{\pm0.195}$ | $0.671_{\pm0.137}$ |
| cX-r05 | $0.609_{\pm0.031}$ | $0.575_{\pm0.193}$ | $0.991_{\pm0.002}$ | $0.999_{\pm0.259}$ | $0.782_{\pm0.089}$ | $0.860_{\pm0.310}$ | $0.840_{\pm0.049}$ | $0.985_{\pm0.126}$ | $0.640_{\pm0.343}$ | $0.714_{\pm0.297}$ |
| cX-r06 | $0.700_{\pm0.014}$ | $0.682_{\pm0.188}$ | $0.988_{\pm0.014}$ | $0.979_{\pm0.250}$ | $0.829_{\pm0.055}$ | $0.976_{\pm0.224}$ | $0.839_{\pm0.064}$ | $1.021_{\pm0.168}$ | $0.723_{\pm0.204}$ | $0.701_{\pm0.171}$ |
| cX-r07 | $0.785_{\pm0.029}$ | $0.773_{\pm0.146}$ | $0.988_{\pm0.021}$ | $0.949_{\pm0.309}$ | $0.813_{\pm0.054}$ | $0.868_{\pm0.194}$ | $0.847_{\pm0.080}$ | $0.983_{\pm0.328}$ | $0.732_{\pm0.097}$ | $0.721_{\pm0.245}$ |
| cX-r08 | $0.863_{\pm0.031}$ | $0.847_{\pm0.184}$ | $0.992_{\pm0.017}$ | $0.966_{\pm0.206}$ | $0.776_{\pm0.076}$ | $0.698_{\pm0.278}$ | $0.851_{\pm0.060}$ | $1.009_{\pm0.113}$ | $0.717_{\pm0.068}$ | $0.764_{\pm0.303}$ |
| cX-r09 | $0.889_{\pm0.041}$ | $0.838_{\pm0.164}$ | $0.996_{\pm0.009}$ | $0.957_{\pm0.443}$ | $0.805_{\pm0.030}$ | $0.750_{\pm0.135}$ | $0.856_{\pm0.057}$ | $1.022_{\pm0.084}$ | $0.729_{\pm0.063}$ | $0.694_{\pm0.298}$ |
| cX-r10 | $0.874_{\pm0.054}$ | $0.862_{\pm0.251}$ | $0.993_{\pm0.014}$ | $0.956_{\pm0.587}$ | $0.777_{\pm0.028}$ | $0.638_{\pm0.147}$ | $0.864_{\pm0.043}$ | $1.023_{\pm0.213}$ | $0.731_{\pm0.082}$ | $0.746_{\pm0.300}$ |
| c01-rX | $0.756_{\pm0.054}$ | $0.161_{\pm0.123}$ | $0.675_{\pm0.322}$ | $0.000_{\pm0.080}$ | $0.681_{\pm0.064}$ | $0.029_{\pm0.010}$ | $0.718_{\pm0.007}$ | $0.165_{\pm0.144}$ | $0.579_{\pm0.036}$ | $0.003_{\pm0.003}$ |
| c02-rX | $0.800_{\pm0.054}$ | $0.239_{\pm0.103}$ | $0.990_{\pm0.006}$ | $0.083_{\pm0.097}$ | $0.726_{\pm0.022}$ | $0.095_{\pm0.014}$ | $0.765_{\pm0.033}$ | $0.238_{\pm0.037}$ | $0.579_{\pm0.014}$ | $0.000_{\pm0.000}$ |
| c03-rX | $0.815_{\pm0.049}$ | $0.264_{\pm0.111}$ | $0.992_{\pm0.011}$ | $0.148_{\pm0.032}$ | $0.734_{\pm0.017}$ | $0.194_{\pm0.073}$ | $0.810_{\pm0.070}$ | $0.332_{\pm0.076}$ | $0.668_{\pm0.095}$ | $0.191_{\pm0.191}$ |
| c04-rX | $0.843_{\pm0.071}$ | $0.380_{\pm0.195}$ | $0.995_{\pm0.020}$ | $0.247_{\pm0.113}$ | $0.740_{\pm0.054}$ | $0.301_{\pm0.178}$ | $0.861_{\pm0.069}$ | $0.413_{\pm0.104}$ | $0.699_{\pm0.028}$ | $0.250_{\pm0.150}$ |
| c05-rX | $0.862_{\pm0.064}$ | $0.431_{\pm0.194}$ | $0.998_{\pm0.006}$ | $0.373_{\pm0.167}$ | $0.748_{\pm0.054}$ | $0.387_{\pm0.198}$ | $0.864_{\pm0.070}$ | $0.468_{\pm0.115}$ | $0.707_{\pm0.073}$ | $0.375_{\pm0.175}$ |
| c06-rX | $0.864_{\pm0.065}$ | $0.511_{\pm0.139}$ | $0.997_{\pm0.006}$ | $0.472_{\pm0.148}$ | $0.765_{\pm0.029}$ | $0.446_{\pm0.330}$ | $0.870_{\pm0.058}$ | $0.499_{\pm0.126}$ | $0.738_{\pm0.077}$ | $0.411_{\pm0.111}$ |
| c07-rX | $0.893_{\pm0.043}$ | $0.580_{\pm0.178}$ | $0.993_{\pm0.013}$ | $0.445_{\pm0.115}$ | $0.773_{\pm0.029}$ | $0.497_{\pm0.370}$ | $0.846_{\pm0.066}$ | $0.483_{\pm0.175}$ | $0.748_{\pm0.097}$ | $0.528_{\pm0.128}$ |
| c08-rX | $0.889_{\pm0.049}$ | $0.635_{\pm0.123}$ | $0.997_{\pm0.011}$ | $0.630_{\pm0.190}$ | $0.772_{\pm0.034}$ | $0.481_{\pm0.367}$ | $0.847_{\pm0.059}$ | $0.438_{\pm0.146}$ | $0.747_{\pm0.038}$ | $0.518_{\pm0.118}$ |
| c09-rX | $0.877_{\pm0.069}$ | $0.628_{\pm0.063}$ | $0.993_{\pm0.017}$ | $0.648_{\pm0.192}$ | $0.769_{\pm0.041}$ | $0.495_{\pm0.326}$ | $0.837_{\pm0.061}$ | $0.451_{\pm0.158}$ | $0.754_{\pm0.037}$ | $0.566_{\pm0.167}$ |
| c10-rX | $0.886_{\pm0.052}$ | $0.646_{\pm0.120}$ | $0.995_{\pm0.021}$ | $0.743_{\pm0.117}$ | $0.768_{\pm0.045}$ | $0.490_{\pm0.392}$ | $0.836_{\pm0.049}$ | $0.424_{\pm0.167}$ | $0.750_{\pm0.051}$ | $0.655_{\pm0.155}$ |

