# OpenReview forum: "Safe Decision Transformer with Learning-based Constraints"
_NeurIPS.cc/2024/Workshop/SafeGenAi — SafeGenAi Poster_

### Official Review · Reviewer_vpub · 2024-10-08
**Good paper but maybe less related to the topic**

**Rating:** 6
**Confidence:** 3

**Review:**

### Summary
The paper introduces the Constrained Q-learning Decision Transformer, which is an innovative approach for enhancing policy synthesis in safe offline RL. By integrating Q-learning with decision transformer techniques, CQDT refines the training dataset via a novel trajectory relabeling scheme that leverages learned value functions to recalibrate future reward and cost predictions in trajectory data. This method significantly boosts the ability to synthesize near-optimal policies while adhering to safety constraints. It is indeed a sound paper with interesting insights. However, I am a little bit suspicious about how this paper is relevant to the main topic of the workshop.

### Strengths
- CQDT effectively synthesizes optimal trajectories by stitching suboptimal segments.
- The model demonstrates robust performance across various safety-critical RL benchmarks.
- The trajectory relabeling mechanism, which adjusts the return-to-go (RTG) and cost-to-go (CTG) based on learned value functions, is a novel contribution that improves the data efficiency and policy quality derived from offline datasets.
- Extensive experiments validate the superiority of CQDT over existing methods like CDT and other baseline models in synthesizing safer and more rewarding policies.
- The paper explores the sparse-reward situation with a new dataset, it is a good starting point for the next step!

### Weaknesses
- The integration of Q-learning with decision transformers adds a layer of complexity in implementing the CQDT framework, potentially increasing the difficulty of tuning and maintenance in practical applications. The training time could be also reported in the appendix.
- I am curious about the performance of the relabelling strategy in sparse reward environments, the paper could provide more evidence to show that. Since in dense-reward environments, it is within my expectation that CQDT would perform better than CDT, but most real-life scenarios are more similar to sparse-reward envs. If the model could not find any possible route initially, would it still be able to find the balance between cost and reward?
- There is a risk that the model could overfit the relabeled trajectories, especially if the diversity in the offline dataset is low. This might result in a policy that performs well on the training data but fails to generalize to new or unseen scenarios.

---

### Official Review · Reviewer_BCkN · 2024-10-09
**used Constrains Penalized Q-learning to improve Constrained Decision Transformer**

**Rating:** 7
**Confidence:** 3

**Review:**

The authors use cost and reward critic functions from Constrains Penalized Q-learning to relabel the return-to-go and cost-to-go in training data. They then use the relabeled data to train a constrained decision transformer (CDT). Their experimental results demonstrates that their proposed method outperforms existing benchmarks.

Strength: comprehensive empirical evaluations, including results on various environments and several baselines, which are commonly used in the literature. Both return and safety are evaluated. Additionally, they performed in-depth analysis on stitching-reward ability, under a variety of settings. Both sparse reward and dense reward environments are studied.

Weakness: As the authors mentioned, it would be nice to theoretically justify the effectiveness of the proposed method. Overall, it is a solid paper with thorough empirical evaluations. Providing some theoretical analysis or at least some theoretical insights would strengthen the paper.

Additional comments:
- In table 1, the authors presented normalized reward, where the normalization is done with respect to the maximum and minimum empirical reward for each task. Why is there a negative value in the normalized return (CPQ on DroneCircle)? It would be nice to add a sentence to the paper to briefly explain that.
- What are the technical challenges in applying Constrains Penalized Q-learning to CDT? In the method section is limited, the authors clearly explained the algorithm but provided very limited discussion on difficulties challenges involved in developing the proposed method. It would improve the paper to highlight these.

---

### Official Review · Reviewer_kp3h · 2024-10-10
**Valuable Contribution to Safe Offline RL, but Needs Clarity and Refinement**

**Rating:** 6
**Confidence:** 3

**Review:**

### Summary
The authors propose the Constrained Q-learning Decision Transformer (CQDT), which addresses limitations in existing Transformer-based safe RL methods. CQDT improves upon this by integrating learned value functions for both reward and cost, using a trajectory relabeling scheme. This enables CQDT to stitch together suboptimal trajectories while maintaining safety.
### Strengths
1. **Interesting Approach**: The combination of Constrained Decision Transformer (CDT) with Constrained Penalized Q-learning (CPQ) is not in itself extremely novel, however it addresses key shortcomings in safe offline reinforcement learning by enhancing trajectory stitching while maintaining safety.
2. **Strong Performance**: CQDT demonstrates good performance across multiple safe RL benchmarks, while not violating safety constraints.
3. Good Evaluation**: The paper presents a wide range of experiments, including those in sparse reward settings.
Weaknesses
1. **Figure 2 Clarification**: It is unclear whether the error bars represent confidence intervals or standard deviations. Additionally, the paper does not specify whether these results are averaged across tasks or based on a specific subset.
2. **Higher Costs for CQDT**: Despite CQDT's improved performance in terms of rewards, it often incurs higher costs than CDT. The paper does not sufficiently address why CQDT leads to higher costs and if this is a trade-off inherent to its design.
3. **Notation and Clarity Issues**: Some parts of the paper, especially regarding the notation, are hard to follow. In some parts of the paper, it is also necessary to read the appendix, which makes it difficult for readers to fully understand the method without diving into additional materials.
4. **X-to-Go Definition:** The reward-to-go and cost-to-go definitions are incorrect. They should read $R_{t} = \sum_{i=t}^T r_{i}$. Similarly for the cost.
5. **Observations $o_t$:** The elements of the observations should be $R_{t-K : t}$, instead of $R_{-K : t}$.
6. **CVAE Training Unclear**: The paper does not provide sufficient details on how the Conditional Variational Autoencoder (CVAE) is trained to predict 0 and 1.
7. **Discount Factor Confusion**: The use of the discount factor γ\gammaγ in the Q-learning updates is inconsistent with the finite horizon, undiscounted formulation introduced earlier in the paper.
8. **Artificial Sparse Reward Setting**: The sparse reward setting in the experiments seems artificially constructed by giving rewards every 10 steps. It would be useful to explore what happens if rewards are only given at the end of an episode, as this would reflect more realistic scenarios.
9. **Cost of Training for Different Thresholds**: Training separate Q-functions for multiple cost thresholds $\kappa$ seems computationally expensive. This overhead is not discussed in detail.
10. **Higher Cost for CQDT**: As shown in Table 1, CQDT frequently incurs higher costs than CDT, often exceeding the safety threshold. The paper does not provide a satisfactory explanation for this behavior.
11. **Limitations Not Discussed**: The paper does not sufficiently discuss its limitations.
### Questions:
1. **Confidence Intervals vs Standard Deviation**: Are the error bars in Figure 2 confidence intervals or standard deviations? What tasks are these results averaged over?
2. **Cost Increase for CQDT**: Why does CQDT tend to incur higher costs compared to CDT? Is there an inherent trade-off between reward maximization and maintaining lower costs?
3. **Sparse Reward Experiment**: What happens if sparse rewards are only given at the end of the episode instead of every 10 steps? How would this impact the stitching ability of CQDT?
4. **Gamma in Q-Learning**: Why was the discount factor $\gamma$ used in the Q-learning updates when the problem was initially formulated as a finite-horizon, undiscounted MDP?
5. **Training Efficiency**: How costly is it to train the Q-functions for different cost thresholds $\kappa$ in terms of time and computational resources? Is this feasible for large-scale problems?